# The evolution of menopause in toothed whales

Samuel Ellis[1]✉, Daniel W. Franks[2], Mia Lybkær Kronborg Nielsen[1], Michael N. Weiss[1,3] & Darren P. Croft[1,3]

Understanding how and why menopause has evolved is a long-standing challenge across disciplines. Females can typically maximize their reproductive success by reproducing for the whole of their adult life. In humans, however, women cease reproduction several decades before the end of their natural lifespan[1,2]. Although progress has been made in understanding the adaptive value of menopause in humans[3,4], the generality of these findings remains unclear. Toothed whales are the only mammal taxon in which menopause has evolved several times[5], providing a unique opportunity to test the theories of how and why menopause evolves in a comparative context. Here, we assemble and analyse a comparative database to test competing evolutionary hypotheses. We find that menopause evolved in toothed whales by females extending their lifespan without increasing their reproductive lifespan, as predicted by the 'live-long' hypotheses. We further show that menopause results in females increasing their opportunity for intergenerational help by increasing their lifespan overlap with their grandoffspring and offspring without increasing their reproductive overlap with their daughters. Our results provide an informative comparison for the evolution of human life history and demonstrate that the same pathway that led to menopause in humans can also explain the evolution of menopause in toothed whales.

Prolonged female postreproductive life—hereafter menopause—is an evolutionary oddity that has puzzled biologists and anthropologists for decades. In human societies, women living under natural survival and fertility can expect to spend an average of 42.5% of their adult life postreproductive[1]. This contrasts starkly with our closest extant relatives: in most wild populations, female chimpanzees spend on average only 2% of their adult life postreproductive[1] (but see ref. 6). Menopause is a very rare phenomena and humans are the only terrestrial mammals demonstrated to have evolved an extended female postreproductive lifespan under natural conditions[5]. This rarity is perhaps unsurprising: under most circumstances, individuals can maximize their fitness by continuing to reproduce for their entire adult lifespan.

There is evidence that menopause is adaptive in humans[2,3]. Although the pathways and mechanisms by which menopause evolves remain debated, the most well-studied and well-supported theories focus on kin selection[3] and the balance of evidence suggests that menopause in humans evolved because of a combination of the benefits of intergenerational help and the need to avoid costly intergenerational harm[3]. Supporting this interpretation, research has found that the presence of mothers and grandmothers can increase the survival of their offspring and grandoffspring[7] and there is evidence that ceasing reproduction can allow older females to avoid costly competition with their relatives[8].

Research on the evolutionary history of menopause has been constrained because menopause is a rare taxonomic trait in wild populations. In primates, for example, the scope for robust comparisons is limited because menopause has only evolved once and humans are unusual among primates in key aspects of their ecology and life history. The recent demonstration of the repeated evolution of menopause in toothed whales (suborder Odontoceti) provides a unique opportunity to investigate the evolution of menopause in a comparative context[9]. In contrast to primates, in toothed whales, menopause has independently evolved at least four times across five species (Fig. 1a), once in the branch leading to short-finned pilot whales (*Globicephala macrorhynchus*[10]), false killer whales (*Pseudorca crassidens*[11]) and killer whales (*Orcinus orca*[12,13]) and once in the branch leading to narwhals (*Monodon monoceros*[9]) and beluga whales (*Delphinapterus leucas*[9]). In this study, we leverage this repeated evolution of menopause to determine whether menopause in toothed whales evolved through the lengthening of the total lifespan or by shortening of the reproductive lifespan (the live-long or stop-early hypotheses). We then evaluate inclusive fitness explanations for menopause by investigating whether this life history strategy increases the opportunity for intergenerational help while minimizing intergenerational harm. Taken together, our findings provide a unique opportunity to evaluate the generality of the theoretical frameworks proposed to explain the evolution of menopause.

## Live-long or stop-early?

In humans, two pathways have been proposed for how menopause evolved: the live-long hypothesis and the stop-early hypothesis[4,14,15]. The

[1]Centre for Research in Animal Behaviour, Department of Psychology, University of Exeter, Exeter, UK. [2]Department of Biology, University of York, York, UK. [3]Center for Whale Research, Friday Harbor, WA, USA. ✉e-mail: s.ellis@exeter.ac.uk

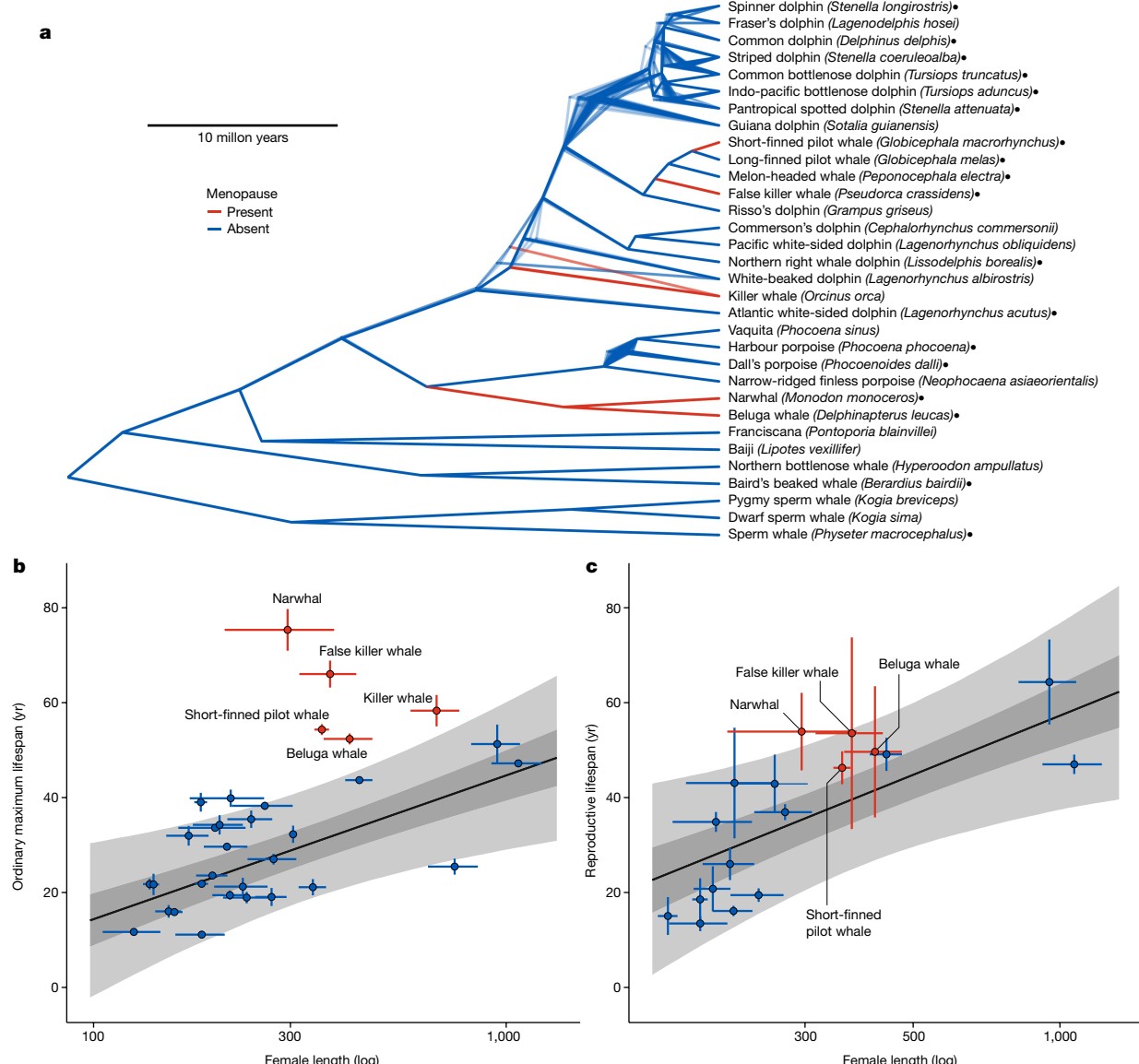

**Fig. 1 | The evolution of menopause in toothed whales. a**, Chronological trimmed phylogenies of Odontocete cetaceans (derived from ref. 35) showing species with lifespan data calculated by and included in this study. The phylogeny shows 1,000 overlapping alternative tree structures. The four independent transitions to menopause are shown with red branches and all branches without menopause are shown in blue. Species names marked with a filled point are those for which we also have estimates of reproductive lifespan derived from ovarian corpora. **b**, The relationship between body length and ordinary maximum lifespan (age at which 90% of adult years have been lived) in female toothed whales with lifespan data (*n* = 32). Points (red with menopause; blue without menopause) show the estimated mean value, with error (s.d.) in both the size and lifespan dimensions; this error is carried through the analysis (all points labelled: Supplementary Fig. 1a). The line shows the predicted relationship between body length and lifespan in whales without menopause (posterior mean ± 50% CI (darker ribbon) and ±95% CI (lighter ribbon)). Species with menopause have longer lifespans than expected given their size. **c**, The relationship between reproductive lifespan and body length in toothed whale species with reproductive lifespan data (*n* = 18, filled points in **a**). Points, lines and ribbons as for **b**. Reproductive lifespan is the age at which ovarian activity is predicted to cease. Species with menopause (red) do not have shorter reproductive lifespans than expected given their size (all points labelled: Supplementary Fig. 1b).

live-long hypothesis proposes that menopause evolved by increasing the total lifespan while the reproductive lifespan remained unchanged from the non-menopausal ancestor, which leads to the comparative prediction that species with menopause will have a longer total lifespan but the same reproductive lifespan as species without menopause. In contrast, the stop-early hypothesis argues that menopause evolved by shortening the reproductive lifespan while the total lifespan remained unchanged from the non-menopausal ancestor leading to the comparative prediction that species with menopause will have the same lifespan as, but a shorter reproductive lifespan than, species without menopause. Comparative work between humans and other primates

provides some support for the live-long hypothesis: in humans, women have a longer lifespan but the same reproductive lifespan as female chimpanzees[16–18]; and the same reproductive lifespan as expected for a primate of their size[1,19].

To test the predictions of the live-long and stop-early hypothesis in toothed whales, we collated and analysed a multispecies database of toothed whale life history. We developed and applied Bayesian mortality models to all available sex-specific age-structured Odontocete data, resulting in female lifespan estimates for 32 species of toothed whale with representatives from all the main clades (Fig. 1a). The models estimate the parameters of a Gompertz mortality model

from the ages of deceased whales (for example, from mass-stranding events), while capturing uncertainty introduced by population growth, sampling biases and errors in age estimation. We estimated reproductive lifespan by using age-linked ovarian corpora counts from whale postmortems as indicators of ovarian activity[9]. We fit models to calculate species-specific changes in ovarian activity with age and define reproductive lifespan as the age at which ovarian activity ceases. Using all available toothed whale corpora data we estimate the reproductive lifespan of 18 species of toothed whale, including four of the five species with menopause (Fig. 1a).

In a phylogenetically controlled analysis, we found that species with menopause live longer than expected given their size (Fig. 1b; proportion of posterior greater than zero is 0.99). The ordinary maximum lifespan (age at which 90% of adult life years have been lived) of whales with menopause is predicted to be 40 ± 5 yr (mean ± s.d.) longer than the same-sized species without menopause. In contrast, species with menopause do not have a shorter reproductive lifespan than expected given their size (Fig. 1c; proportion of posterior less than zero is 0.20). These results are consistent with the predictions of the live-long hypothesis but not the stop-early hypothesis.

Taken together, these results demonstrate that in toothed whales, just as in humans, menopause evolved by species extending their lifespan without a corresponding increase in reproductive lifespan. Our finding that menopause in toothed whales evolves by selection acting to extend lifespan is consistent with evolutionary theories of ageing, arguing that the costs and constraints of somatic and reproductive ageing are likely to be different and that, under the right social conditions, it may be easier for lifespan than for reproductive lifespan to be extended[20]. Theories for the evolution of menopause, therefore, need to establish the social conditions for (1) why lifespans have extended and (2) why reproductive lifespan is not also extended.

## Help and extended lifespans

Intergenerational help has been identified as a key adaptive benefit of living long in species with menopause. As in humans, in all toothed whale species with menopause for which the social structure is known the whales inhabit multigenerational kin-structured groups and therefore have the potential for important roles of intergenerational help and harm in their life history. In humans and killer whales, grandmothers increase the survival probability of their offspring and grandoffspring[7,21,22]. It has been further argued that providing intergenerational help—particularly help from grandmothers to grandoffspring—is the selection pressure that pushed females to extend their lifespan[4,23,24]. It follows that species with menopause are expected to spend more time alive at the same time as younger generations than species without menopause.

We applied kinship demography models[25] to Odontocete life history data to calculate the potential for intergenerational help in toothed whales with and without menopause. We parametrized the kinship demography models using the adult age-specific survival and fecundity measures developed for our live-long and stop-early analyses, as well as estimates of juvenile survival derived from further Bayesian modelling of whale mortality data. We characterized the potential for intergenerational help in each whale species as the expected total number of years an adult female spends alive at the same time as her grandoffspring and offspring. We calculate offspring and grandoffspring years at a given age as the number of offspring/grandoffspring a female can expect to have at that age multiplied by the probability of a female surviving to that age. We then sum this metric over all ages to get the total offspring/grandoffspring years for that species. To compare between species we derive 'relative' offspring and grandoffspring years by dividing the total number of years by the species age at maturity. We also extrapolated from extant species to simulate the demographic parameters and hence the opportunity for intergenerational help, of

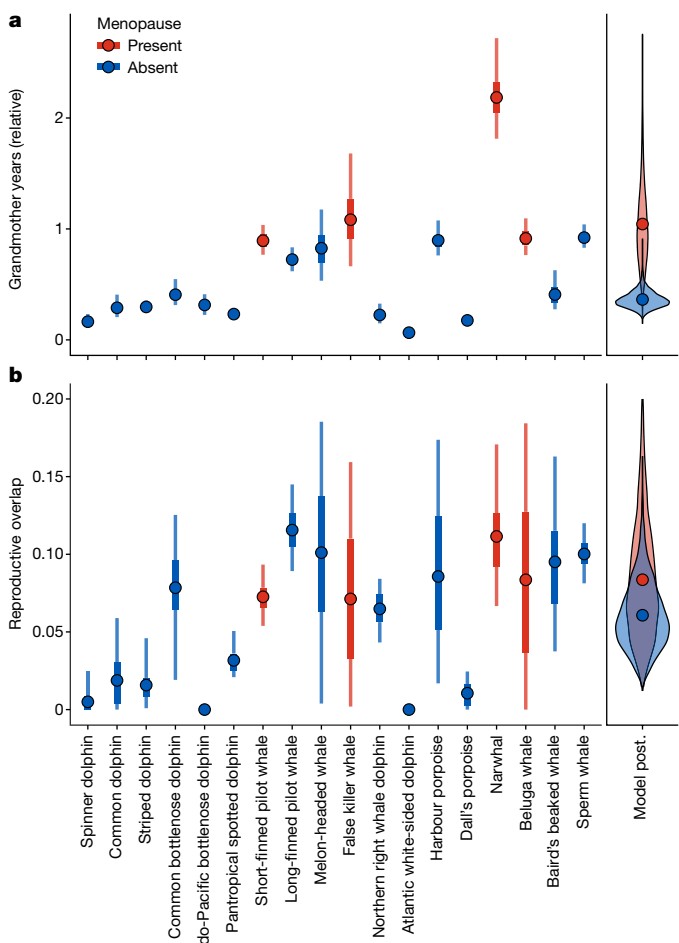

**Fig. 2 | Opportunities for intergenerational help and harm in toothed whales. a,b**, Relative grandmother years (**a**) and mother–daughter reproductive overlap (**b**) for toothed whale species ($n$ = 18; Fig. 1a 'filled point' species) with (red) and without (blue) menopause. Relative grandmother years represent the number of grandoffspring years an adult female can expect to experience, scaled to the species-specific age at maturity. Reproductive overlap is the summed proportion of adult female reproduction that occurs concurrently with the reproduction of a daughter. The left-hand panels show estimates of both grandmother years and reproductive overlap and are derived from 1,000 kinship demography models with parameters derived from the posterior distributions of the lifespan and reproductive lifespan models applied to each species. Points in this panel show the mean estimate of the parameter, whereas the thick and thin error bars show the 50% and 95% CIs, respectively. The right-hand panels show the distribution of posterior estimates from a Bayesian model (Model post.) estimating the mean demographic parameter expected over all species with and without menopause; points show the mean estimate. Species with menopause have more grandmother years but the same reproductive overlap as species without menopause.

the ancestors (shorter lifespan, same reproductive lifespan) of species with menopause.

Species with menopause spend longer alive at the same time as their grandoffspring than species without menopause (Fig. 2; $\beta$ = 1.01, 95% credible interval (CI) = 0.31–1.67; proportion of posterior menopause greater than no menopause is 0.99). In species with menopause grandmothers are predicted to live 1.04 ± 0.38 relative grandoffspring years (grandoffspring years/age at maturity: units for a given species are therefore 'age at maturities') with their grandoffspring, whereas in species without menopause grandmothers are only expected to live 0.36 ± 0.08 relative grandmother years with their grandoffspring. Put another way, species with menopause can expect to live long enough

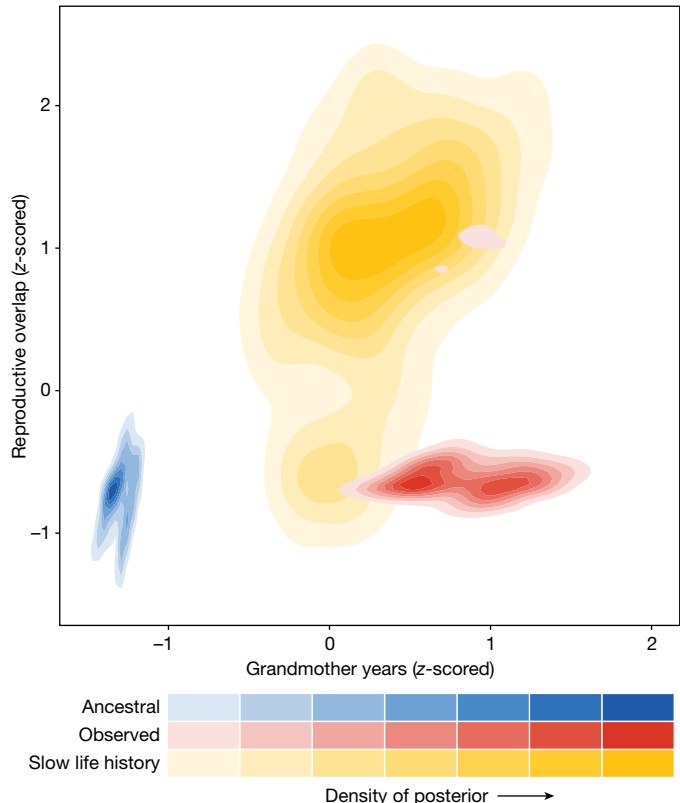

**Fig. 3 | Help and harm implications of life history strategies for species with menopause.** Comparing the estimated relative grandmother years and reproductive overlap for all five toothed whale species with menopause as: observed in real systems (observed, red areas); in their simulated ancestral non-menopausal state (ancestral, blue areas); and a simulated analogue of the species that keeps reproducing to the end of life (slow life history, yellow areas). Contoured areas show the distribution of the posterior distributions under each state. Kinship demographic models are run separately for each species-state but, for illustration, posterior estimates have been combined and z-scored in species to get comparable between-species estimates. In the menopause state (red) females have more grandmother years but the same reproductive overlap as their ancestral state (blue) and a lower reproductive overlap than in the slow life history state (yellow).

for more than one grandoffspring to reach maturity, whereas species without menopause can only expect to see an offspring to 36% of age at maturity. Species with menopause, therefore, have a greater opportunity for females to assist in raising their grandoffspring than species without menopause. Supporting this, demographic simulations show that species with menopause have an increased overlap with their grandoffspring compared to their non-menopausal ancestor (Fig. 3, observed versus ancestral cases).

Intergenerational help over one generation (from mothers to their offspring) has also been suggested to be important to the evolution of menopause[14,26]. Supporting this we found that, as with grandoffspring, females in species with menopause live for longer at the same time as their offspring than do species without menopause ($\beta = 0.37$, 95% CI = 0.14–0.59; proportion of posterior no menopause greater than menopause is 0.01). Some theoretical work goes further and predicts that in toothed whales with menopause females should preferentially direct their help at male relatives[27]. In systems in which males mate outside of the local group, their offspring do not contribute to ingroup resource competition: females can therefore increase their inclusive fitness without increasing within-group reproductive competition by investing in increasing the reproductive success of their male relatives[27]. This hypothesis is supported in killer whales: male resident killer whales

receive costly investment from their mothers throughout their life[28]. Assuming that male size reflects competitive ability, this hypothesis leads to the comparative prediction that males will be relatively larger than females in species with menopause than in species without menopause because female investment in males allows them to reach and maintain a larger size, increasing their competitive ability[29]. Consistent with this prediction we find that in species with menopause the female/male adult size ratio is smaller than in species without menopause ($\beta = -0.18$, 95% CI = −0.36 to −0.007; proportion of posterior for which female/male size ratio in species with menopause is less than without menopause is 0.96; Supplementary Fig. 2). That is, males in species with menopause are predicted to be a mean of 20 ± 7.7% larger than females compared to 5 ± 4% larger in species without menopause. This is consistent with the prediction that, in species with menopause, help will be directed at males.

## Help and reproductive lifespans

Intergenerational help has also been proposed as a reason why reproductive lifespan has not extended in species with menopause[23,26]. Some models have shown that when providing intergenerational help is so costly it cannot be provided at the same time as reproducing, selection will act to extend female lifespan without also extending reproductive lifespan[30–32]. These models lead to the comparative prediction that the offspring of species with menopause will be more costly to raise to maturity than offspring without menopause and therefore require costly intergenerational help for reproduction to be successful.

We found no evidence that the offspring of species with menopause require more investment to successfully reach maturity than the offspring of species without menopause. As measures of investment in offspring, we used age at maturity as a metric of the time cost of raising an offspring; and adult body size as a metric of how much growth needs to occur during the juvenile period. Neither females nor males in species with menopause have a later age at maturity than expected for the size of their mothers (females, $\beta = 0.20$, 95% CI = −0.33–0.72; males, $\beta = 0.25$, 95% CI = −0.28–0.76; Supplementary Fig. 3). Similarly, despite males being relatively larger in species with menopause than without menopause (above), overall in neither sex do adults in species with menopause reach a larger size than expected given their sex-specific age at maturity (females, $\beta = 0.18$, 95% CI = −0.33–0.69; males, $\beta = 0.23$, 95% CI = −0.29–0.74; Supplementary Fig. 4).

In some models of intergenerational help and reproductive lifespan, the benefit provided by older females has been modelled as an increased reproductive rate in the daughters of postreproductive females[30,33]. Under this mechanism, the observed reproductive rate of species with menopause is predicted to be higher than in their ancestors without menopause. However, we found no evidence to support this prediction in toothed whales. We calculate baseline reproductive rate as the rate of reproduction necessary to maintain a stable population size given the age-specific mortality trajectories[34] as calculated from our mortality model or predicted for our demographic simulations. Contrary to this hypothesis, in all five species with menopause, the baseline rate of reproduction is predicted to be lower in the observed case than in the ancestral case (proportion of posterior observed greater than ancestral is 1 for all species). This suggests that species with menopause are reproducing more slowly than their non-menopausal ancestor. This result is consistent with research in killer whales demonstrating that females with a living postreproductive mother have a longer interbirth interval than those without a living postreproductive mother[21].

There are, however, a variety of other ways that older females can increase the fitness of their younger kin other than increasing their daughters' reproductive rate, such as increasing the survival of their offspring and grandoffspring or increasing the reproductive success of their sons. Indeed, as discussed above, investment in the success of sons has been proposed as the key pathway by which postreproductive

females provide intergenerational help in killer whales[27,35], which implies that we may not expect the presence of menopause to change daughters' reproductive rate. Further, although here we have focussed on two measures of help before maturity, other measures (for example, age-specific growth rates) would be interesting to analyse as data become available. Intergenerational help can also be provided after the receiver has reached maturity: for example, in resident killer whales, females provide life-long investment to their adult sons at a cost to their own reproductive output[28].

## Harm and reproductive lifespans

Intergenerational harm has been implicated as a cost of extending reproductive lifespan in species with menopause. Reproducing can be considered a form of generalized harm because reproduction increases within-group competition for resources[36]. Age-linked relatedness asymmetries can lead to older females competing less for reproductive resources than younger females because of the costs of inflicting harm on their concurrently breeding younger relatives[36,37]. Studies in humans and killer whales have supported a cost for older females reproducing at the same time as their daughters and daughters-in-law, respectively, in species with menopause[8,37]. Under a live-long framework, not extending the reproductive lifespan while living longer is a way for females to gain the inclusive benefits of intergenerational help while avoiding costly intergenerational conflict. In effect, this intergenerational harm theory argues that menopause allows females to separate their reproductive generations while maintaining a long lifespan overlap[36]. The intergenerational harm theories predict that (1) species with menopause will have the same reproductive overlap as species without menopause, despite living longer and (2) in species with menopause the observed reproductive overlap will be lower than if they had also extended their reproductive lifespan and kept reproducing to the end of their lifespan.

We used the same kinship demography models to measure intergenerational help to characterize the potential for intergenerational harm in toothed whales with and without menopause. The potential for intergenerational harm was measured as reproductive overlap, calculated as the summed proportion of an adult females' reproductive capacity that overlaps with her daughters' reproduction. Specifically, for all female ages we multiply the number of grandoffspring of age 0 they can expect to have at that age by the remaining reproductive capacity from that age. This is then summed over all ages. This measure characterizes how much of a female's lifetime reproduction is expected to occur at the same time as when her daughter is reproducing. We also calculated the opportunity for intergenerational harm in two simulated analogues of the observed species with menopause: an ancestral case (lifespan matching observed reproductive lifespan) and a slow life history case (reproductive lifespan matching observed lifespan).

As predicted, species with menopause do not have a different mother–daughter reproductive overlap than species without menopause (Fig. 2; $\beta$ = 0.27, 95% CI = −0.49–0.97; proportion of posterior menopause greater than no menopause is 0.73). In both species with and without menopause the summed proportion of female reproduction occurring at the same time as her daughter is 0.05–0.09 (50% CI). Supporting this result, we did not find any evidence that a model including menopause as a parameter has greater predictive power than a model without the menopause parameter (expected log pointwise predictive density (elpd) difference without − with = −0.4 ± 1.0). Further, demographic simulations show that in species with menopause the observed reproductive overlap is lower than if they had continued to reproduce for their whole lifespan (Fig. 3, observed versus slow life history cases). The simulations also show that species with menopause have not increased their reproductive overlap compared to their non-menopausal ancestor (Fig. 3, observed versus ancestral cases).

### Table 1 | Summary of hypotheses and results

| Hypotheses | Prediction | Result |
| --- | --- | --- |
| **Live-long hypothesis.** Menopause evolves when lifespan is extended without also extending the reproductive lifespan | In species with menopause, females will live longer than, but have the same reproductive lifespan as, species without menopause, given their size | Supported (Fig. 1) |
| **Stop-early hypothesis.** Menopause evolves when the reproductive lifespan is shortened without changing the lifespan | In species with menopause, females will have the same lifespan as, but a shorter reproductive lifespan than, species without menopause, given their size | Not supported (Fig. 1) |
| **Intergenerational help selects for an extended lifespan.** Extended female lifespans in species with menopause evolve to allow older females to increase the fitness of their offspring and grandoffspring | Females in species with menopause will spend longer alive at the same time as their grandoffspring and offspring than females in species without menopause or their non-menopausal ancestors | Supported (Figs. 2 and 3) |
| **Intergenerational help selects against an extended reproductive lifespan.** Reproductive lifespan is not extended in species with menopause because they can gain higher fitness by helping younger generations to reproduce | Reproduction in species with menopause will be more costly than in species without menopause. And the rate of reproduction will be higher in species with menopause than in their ancestor without menopause | Not supported |
| **Intergenerational harm and reproductive lifespan.** Reproductive lifespan is not extended in species with menopause because of the inclusive fitness costs of reproducing at the same time as offspring | Females in species with menopause will have the same reproductive overlap as species without menopause and their non-menopausal ancestor despite living longer | Supported (Figs. 2 and 3) |
| **Males and the evolution of menopause.** Female lifespan is extended beyond their reproductive lifespan in species with menopause because of selection on extended male longevity also affecting female lifespan | Males will have relatively longer lifespans in species with menopause than in species without menopause and in species with menopause males will have a higher probability of living to, and will live longer beyond (6.3), the age of female reproductive cessation | Not supported |

## Female lifespans and male longevity

An alternative hypothesis for the evolution of an extended female postreproductive lifespan is the male-driven menopause hypothesis, which argues that female menopause is an artefact of selection on extended male lifespan[38–40]. Were female menopause an artefact of selection on male lifespan we would expect that (1) males would live longer in species with menopause than in species without menopause and (2) in species with menopause males will have a higher probability of reaching, and live longer beyond, the age of female reproductive cessation than females (Table 1). We find no evidence to support the male-driven menopause hypothesis in toothed whales. In contrast to the expectations of the male-driven menopause hypothesis, in toothed whales we find that (1) the ratio of female to male lifespan is lower in species with menopause than species without menopause ($\beta$ = 0.186, 95% CI = −0.06–0.31; proportion of posterior below 0 is 0.009) and (2) that in species with menopause females are both more likely to survive to the age of female menopause (for all species, proportion of posterior in which males are more likely to live to female age of menopause than females is less than 0.04) and live longer once they reach

that age (proportion of posterior in which males live longer than females: beluga whale 0.20, narwhal 0.14, all other species less than 0.03) than males. In other words, males in species with menopause have shorter lifespans relative to females than in species without menopause and in species with menopause males are less likely to reach the age of female reproductive cessation than females and live less long from that age if they do reach that age.

## Discussion

We have shown that the evolution of menopause in toothed whales has a striking similarity to the evolution of menopause in humans, with menopause evolving in both humans and toothed whales through the evolution of a longer overall lifespan without concurrently extending the reproductive lifespan (Table 1). We also found that the demographic consequences of female life history can give some insight into the selective pressures driving the evolution of menopause (Table 1). Namely, we found that, compared to species without menopause, in toothed whale species with menopause, females spend more time alive with their grandoffspring but no longer reproducing at the same time as their daughters. These life history similarities suggest important roles for aspects of intergenerational help and intergenerational harm in the evolution of menopause in toothed whales.

The results presented here provide support for a synthesis of different theories for the evolution of menopause. Our comparative analyses of toothed whales are consistent with a key role of grandmother benefits in the evolution of menopause, as predicted by the grandmother hypothesis[4,14,23,24,41,42]. But unlike the expectations of the grandmother and mother hypotheses[14,26], we find no evidence that the benefits are necessary because offspring are particularly costly to raise to maturity in species with menopause or result in a faster baseline rate of reproduction in species with menopause[4,26]. Rather, and consistent with the predictions of the reproductive conflict hypothesis, we find that species with menopause do not have increased reproductive overlap with their offspring, despite their extended lifespan[27,36]. More generally, we find support for the hypothesis that menopause allows females to maintain separate reproductive generations while increasing overall generational overlap. In agreement with some other recent studies in humans[43–45], we find no evidence for the male-driven menopause hypothesis: in toothed whales the evolution of menopause seems to be driven by selection on female life history.

There are various ways that older female toothed whales may provide intergenerational help. Older females can, for example, provide intergenerational help by directly sharing food with their relatives[46], babysitting their grandoffspring[47] or using their ecological knowledge to lead their group when resources are scarce[48]. Older females can and do provide intergenerational benefits in at least some toothed whale species. The effect of intergenerational help has been demonstrated in resident-ecotype killer whales in which the death of an older female increases the mortality risk of both their grandoffspring[21] and particularly their adult sons[22].

Providing intergenerational help, and inflicting intergenerational harm, relies on older females associating with their offspring and grandoffspring. Long-term behavioural studies of killer whales have shown that mothers and their adult offspring of both sexes associate for at least part of[49], and in some ecotypes the entirety of, the mothers' life[50]. Although less definitive, there is growing behavioural and genetic evidence that both false killer whales and short-finned pilot whales inhabit stable mixed-sex social groups consisting of close relatives, including mothers and their adult offspring of both sexes[51,52]. The social structure of beluga whales and narwhals is usually referred to as 'matrifocal' but remains largely unknown. Both species exhibit mitochondrial substructuring in the summer feeding grounds[53,54]. Beluga whales of both sexes are found with close relatives in at least some areas and contexts[55,56] and female beluga exhibit behaviours

consistent with being in close association with relatives[57]. There is, therefore, the potential for long-term mother–offspring associations in all five toothed whale species with menopause and therefore the opportunity for postreproductive females to provide benefits to their offspring and grandoffspring.

Intergenerational help alone, however, cannot explain the evolution of menopause and grandmothers have been reported to enhance grandoffspring survival across a wide range of species that lack a postreproductive lifespan[58]. Indeed, our results indicate that in several species without menopause there is considerable lifespan overlap between grandmothers and their grandoffspring, suggesting the potential for intergenerational help in these species. Menopause will only evolve when both the benefits that older females can provide their local kin are large enough to select for an extended lifespan and the costs of late-life reproduction, for example, because of the consequences of reproductive conflict, are high enough to select against extended reproduction. Kinship dynamics—age-dependent changes in patterns of local relatedness driven by patterns of dispersal and mating[59]—are suggested to play a key role in tipping the scale against late-life reproduction[27]. Because female relatedness to her local group increases as she ages, older females are under strong selection to help and reduced selection to harm, which can favour the evolution of menopause[27]. In contrast, under the same patterns of kinship dynamics, young females are under stronger selection to harm and weaker selection to help[27]. These asymmetries in local relatedness mean that older females can be outcompeted by younger females when they try to reproduce at the same time[37]. Resident-ecotype killer whales exhibit bisexual philopatry with out-group mating leading to increasing female local relatedness with age[59] and, as predicted on the basis of patterns of kinship dynamics, old females lose out in intergenerational reproductive competition with their daughters[37]. Similarly, social structures driving comparable patterns of kinship dynamics may have been present in early Hominids[59]. We predict similar patterns of increasing female relatedness with age in the other toothed whale species with menopause, assuming that current sociality reflects the ancestral social states. Kinship dynamics alone, however, are not sufficient to explain the evolution of menopause. Increasing female relatedness with age is also a feature of species without menopause (for example, long-finned pilot whales[60]). However, despite this kinship dynamics pattern, the mortality rate-doubling time of long-finned pilot whales is half that of short-finned pilot whales, reflecting the reduced importance of, and a reduced opportunity for, late-life selection in species without menopause[61]. For menopause to evolve, the direct fitness costs of ceasing reproduction must be balanced by very high indirect gains that will involve both kinship dynamics relatedness asymmetries and behaviours and ecological circumstances that allow older females to provide large benefits to their kin.

Here, we have found evidence for the roles of intergenerational help and harm in the evolution of menopause in toothed whales. Other selective mechanisms, for example multilevel selection or intergroup conflict, could also be playing roles in the evolution of menopause alongside the kin-selected mechanism highlighted in this study. Further research comparing between toothed whale species or comparing toothed whales to other species in which menopause seems to occur in some populations (for example, chimpanzees in Ngogo[6]) or in semi-natural conditions (for example, Asian elephants *Elephas maximus*[62]) will lead to valuable insights into the evolution of menopause and of life history more generally.

Toothed whales and humans have very different ecologies, modes of sexual competition, sociocultural systems and their last common ancestor lived about 90 million years ago[63]. Despite these differences, our results show that humans and toothed whales show convergent life history. Just as in humans, menopause in toothed whales evolved by selection to increase the total lifespan without also extending their reproductive lifespan. This convergent evolution of menopause

provides important insights for our understanding of the evolution of menopause in general, including in humans.

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

## Methods

For this study we collected, processed and analysed published data from the literature to get estimates of lifespan, reproductive lifespan, size, survival to maturity and age at maturity for as many toothed whale species as possible. We then used these estimates in analyses to understand how and why menopause evolves. All data management, analysis and plotting were performed in R with the tidyverse, rstan cmdstanr and ape packages[64–67].

### Data

**Lifespan data and modelling.** We collected sex-specific age-structured toothed whale data from the literature and then applied a mortality model to these data to get species-sex lifespan estimates for toothed whale species. In most toothed whale species, the age of a whale can be inferred by counting tooth growth rings[68]. There is a tradition in the study of toothed whales to collect age and sex data from deceased individuals, for example, after mass-stranding events. Here, we collate and use these published data to understand toothed whale mortality patterns. We collated the data by performing a systematic search of the literature through Web of Science (webofscience.com) using species name[69], alternative species name[69], common name[69] and alternative common names[69] with each of the search terms 'life history', 'lifespan' and 'age structure'. We also repeated the search using relevant languages local to the distribution of the species in question in an attempt to reduce the English-language bias of our data[70]. We repeated the search for each of the 75 species of toothed whale recognized by the International Whaling Commission[69]. For each species, we also performed opportunistic searches by following references through the literature and identifying potential data sources using the relevant chapters in authoritative edited collections[71–74]. From our search, we identified and extracted[75] sex-specific age data into a database[76]. In the database, where possible, each dataset represents data from a single population collected at a single time; however, this is not always possible and some datasets represent data collected from several populations or over a longer timescale. We also parameterize the mortality model with dataset-specific information on population growth, sampling biases and age-estimation error each of which are based on data provided in the original publication[76]. A species-sex was included in the analysis if they had at least one dataset with more than ten samples above the age of maturity and a sampling rate (number of samples/maximum observed age) of greater than 0.5. Our complete database for analysis used 269 datasets from 118 publications allowing us to estimate lifespan for 32 species of female (Fig. 1 and Supplementary Table 1) and 33 species of male toothed whale. Fisheries by-catch ($n = 113$) and stranding ($n = 107$) are the most common sources of data in our database, although most species have data from several sources. Research to date has not suggested or found evidence of systematic age differences in propensity to be taken as by-catch or stranding and when bias is suspected in a given dataset we include this bias in our modelling framework.

We developed a Bayesian modelling framework to derive the parameters, $\alpha$ and $\beta$, of a Gompertz mortality model for each species-sex. The framework aims to derive a distribution of mortality parameters for each species-sex most parsimonious with the data and with the potential and unknown effects of population growth, sampling bias and age-estimation error. We consider the count $c$ of whales in age category $i$ (where AGE is greater than age at maturity) in dataset $d$ to be drawn from a multinomial distribution, where $\theta_i$ is a product of the probability of surviving to that age, $L_i$, the effect of population change, $R_i$, and the effect of sampling bias, $S_i$ (equation (1)). $L_i$, $R_i$ and $S_i$ are defined on the basis of a function of the age, $AGE_i$, of the whales in category $i$. We concurrently model that the count $c_{d,i}$ is the number of whales with the true age $i$ in dataset $d$, where the true age of whale $j$, $\tau_j$, is drawn from a normal distribution around the observed age, $o_j$, with a standard deviation of $\varepsilon_j$ (equation (1)). The model is fit over all available datasets of the same species. Datasets of the sex ($SEX_d$) share mortality parameters, datasets from the same population ($POP_d$) share a population growth parameter ($r_d$) and each dataset has an independent sampling bias parameter ($s_d$). Sampling bias error is applied where $AGE_{d,i}$ is within a predefined bias window $W_d$ derived from the data source. Simulations demonstrate that this model is capable of recovering reliable estimates of lifespan under a biologically relevant range of demographic and error scenarios[76] (see also Supplementary Information 1, Equations):

$$c \sim \text{Multinomial}(\theta, N)$$

$$\theta_{d,i} = \frac{L_{d,i} R_{d,i} S_{d,i}}{\sum L_{d,i} R_{d,i} S_{d,i}}$$

$$L_i = \frac{e^{-(\alpha_{SEX_d}/\beta_{SEX_d})(e^{\beta_{SEX_d} \times AGE_{d,i}} - 1)}}{\sum_{j=0}^{n} e^{-(\alpha_{SEX_d}/\beta_{SEX_d})(e^{\beta_{SEX_d} \times AGE_{d,j}} - 1)}}$$

$$R_{d,i} = (1 - \rho_d)^{AGE_{d,i}}$$

$$\rho_d = \frac{r_{POP_d}}{1/2 \times \max(o_{DSET_{SEX=SEX_d}})}$$

$$S_{d,i} \sim \begin{cases} s_d + 1 & AGE_{d,i} \in W_d \\ 1 & AGE_{d,i} \notin W_d \end{cases} \quad (1)$$

$$\tau_j \sim \text{Normal}(o_j, \epsilon_j)$$

$$\epsilon_j = \frac{1}{20}(o_j + B)$$

$$c_{d,i} = \sum_{k=1}^{N} [\lfloor \tau_k \rfloor = AGE_i, DSET = d]$$

We fit this using Hamiltonian Monte Carlo chains implemented in R and Stan[64,66] (Supplementary Information 2, Mortality and Corpora). For each species-sex, we then use the posterior estimates of $\alpha$ and $\beta$ to calculate ordinary maximum lifespan (age $Z$): the age at which 90% of adult life years have been lived[77]. This results in a distribution of ordinary maximum lifespans for each species-sex derived from 40,000 draws from the posterior of the fitted mortality model. In the following analyses, we use the mean ± s.d. of the posterior distribution of ordinary maximum lifespans as our species-sex lifespan measures.

**Reproductive lifespan and modelling.** We use age-specific ovarian activity to assess toothed whale reproductive lifespans. In toothed whales, after ovulation, the corpora in which the ovum develops persists in the ovary[78]. A sample of corpora counts from whales of known ages can therefore be used to calculate changes in the rate of corpora deposition with age as a measure of age-specific ovarian activity[9]. In long-finned and short-finned pilot whales the rate of corpora deposition with age matches age-specific pregnancy rates[9]. We use the age at which the rate of new corpora deposition reaches 0 as our measure of reproductive lifespan.

The counts of corpora are commonly reported after dissections of deceased whales. We build on the database of age-specific corpora counts used in previous studies[9] by adding more datasets uncovered during our systematic and opportunistic search for toothed whale life history data (Lifespan data and modelling section). Each corpora dataset consists of a sample of known-age females and the number of ovarian corpora they were found to have. We only include corpora datasets in our analysis if they have a sample size greater than or equal to 20 and a sampling rate (number of samples/maximum observed age from lifespan data) of more than 0.6. Our final sample size for this analysis is 27 datasets from 18 species (Fig. 1, Supplementary Information 2, Mortality and Corpora and Supplementary Table 2).

We apply a Bayesian model to these corpora data to find the age at which ovarian activity reaches 0. Unlike in previous studies[9], here we directly model the process of corpora deposition. We model the count

of corpora at a given adult age $C_i$ (where $i = 0$ is the age at maturity) to be drawn from a Poisson distribution with mean $\lambda_i$, where $\lambda_i$ is the sum of corpora ($k$) deposited at all previous ages, which in turn depends on the initial rate of corpora deposition $\alpha$ and a linear rate of decline in rate with age $\beta$ (see also Supplementary Information 1, Equations):

$$C \sim \text{Poisson}(\lambda)$$
$$\lambda_i = \sum_{j=0}^{i} \Delta k_j \qquad (2)$$
$$\Delta k_j = \alpha(1 - \beta \text{AGE}_j)$$

If several corpora datasets are available from the same species, they are modelled together with datasets allowed to differ in $\alpha$ but sharing an age-specific rate of decline $\beta$. For all species, reproductive lifespan is then calculated as $1/\beta$ + age at maturity. We calculate a distribution of reproductive lifespans from 40,000 draws from the posterior distribution of $\beta$.

**Size and maturity.** We use length rather than mass as our measure of species size because of the difficulty of accurately measuring mass in cetaceans[79]. For each toothed whale species-sex, we consulted expert-written edited volumes[71–74] to get all available metrics of sex-specific length. For each species-sex, we collated all available measures of mean, asymptotic, minimum and maximum length, as well as standard deviation around the mean or asymptote. If metrics were not available from a sex-specific sample ($n = 20$) we instead used any available combined sex samples; if this occurred, the species were not included in any within-species-sex size comparisons. We then processed these metrics to get a single estimate of mean size and standard deviation around that estimated size (Supplementary Information 3, Additional Data Explanation). This error was carried through all subsequent analyses. Available estimates of mass were strongly positively correlated with length (Supplementary Information 3, Additional Data Explanation).

We use the same framework to generate estimates of age at maturity for each species-sex as we did for size (Supplementary Information 3, Additional Data Explanation). We gathered estimates of mean age of maturity and distribution from expert consensus and then processed these measures to get a single measure of mean age of sexual maturity and standard deviation around the mean for each species-sex. For simplicity, for analyses not directly testing correlates of the age of maturity we use the mean measure as the age of sexual maturity.

In analyses regressing size against age or vice versa, models take the form of a Bayesian phylogenetically controlled linear model with the required parameter and menopause as predictors. In these models, both the true size and true age at maturity are considered to be drawn from a normal distribution around the observed mean given the observed standard deviation. The effect of phylogeny is implemented as the covariance between species by means of the Ornestain–Uhlenbeck process[80].

Our kinship demography analysis requires estimates of juvenile mortality. We generate estimates of juvenile mortality for the 23 datasets with good sampling (>100) of whales under the age of maturity by applying a Gompertz mortality model with a bathtub term[81] (to describe juvenile mortality) to these datasets. In these samples, we found a negative correlation between age at maturity and the probability of surviving to maturity. We used this correlation to generate a posterior distribution of the estimated proportion of whales born surviving to maturity for each species-sex (Supplementary Information 3, Additional Data Explanation).

**Kinship demography analysis.** We use a matrix population modelling framework to understand how many relatives of different classes a female can expect to have given her age[25] and hence her potential to provide intergenerational help and intergenerational harm. For each species, we build a Leslie matrix based on (1) mortality hazard at each adult age derived from the lifespan modelling, (2) mortality hazard at juvenile ages derived from the survival to maturity analysis and (3) age-specific fecundity derived from the corpora analysis $\beta$ scaled by the baseline fecundity $f$ needed to maintain a stable population (the age-distribution of a stable population is calculated by inferring the posterior distribution of mortality parameters when $r = 1$ for each species from our fitted mortality models). For each $\beta$ draw from the corpora analysis, we calculated the baseline reproductive rate by systematically exploring the parameter space to find the value of $f$ needed to maintain this known stable age-distribution ($\lambda = 1$) using established matrix population methods[34]. All of these inputs are the posterior estimates from Bayesian analyses. Computational limitations mean that it was not possible to explore the entire combined posterior space, we therefore take 1,000 draws from each posterior distribution and use these to create a set of alternative Leslie matrices for each species. We apply kinship demography models[25]–adapted to predict both sexes of offspring and grandoffspring–to each Leslie matrix to calculate a distribution of the number of offspring and grandoffspring a female can expect to have at a given age.

We quantify grandmother overlap at age $x$ as the number of matrilineal grandoffspring below the age at maturity that a female can expect to have at that age multiplied by the probability of surviving to that age from maturity. We then sum this over all female ages. This measure gives a direct measure of the number of years a female can expect to be alive at the same time as her grandoffspring. Similarly, mother–offspring overlap is quantified as the number of years a female can expect to spend alive at the same time as her offspring and is calculated in the same way as grandmother overlap but replacing the number of grandoffspring with the number of offspring (of both sexes). We measure reproductive overlap as the cumulative proportion of a female's reproductive life remaining when her daughter gives birth. To calculate this at each age we take the product of the expected number of grandoffspring of age 0 that a female will have and multiply this by the proportion of reproductive capacity remaining to a female of that age. This is summed over all ages to get the total reproductive overlap in the species. We use the same methodological pathway to calculate 'relative offspring overlap' between mothers and the offspring, determined separately for daughters and sons.

**Demographic simulations.** For each of the five species with menopause, we also calculated the expected grandmother years and reproductive overlap under two alternative demographic scenarios: (1) the ancestral case and (2) the slow life history case. The ancestral case is the predicted demographics of the immediate non-menopausal ancestors of the species. Under the live-long hypothesis (see the section 'Live-long or stop-early?'), species without menopause evolve by extending their lifespan without extending their reproductive lifespan. For the ancestral case, we therefore simulate the demographic parameters of a population with the same lifespan as expected for the size of the species and the same reproductive lifespan as observed in the real data (see Supplementary Table 3 for the derivation of parameters). The slow life history case simulates a population in which, instead of evolving menopause, females continue to reproduce for their whole life. Under the slow life history case, lifespan is the same as observed for the real species but reproductive lifespan is extended to lifespan (Supplementary Table 3). We compare the ancestral case and slow life history case to the observed demographics (observed case). For this analysis, we measure reproductive lifespan as the age of the oldest known reproductive active female (Supplementary Table 4) to allow the inclusion of killer whales in the analysis, as no corpora data are available for killer whales. Unlike most of the other species in our dataset, the reproductive datasets for these five datasets are large enough to allow a relatively robust estimate of the age of the oldest reproductively

active female. For each demographic scenario, we use the pipeline for the kinship demography analysis (above) to calculate distributions of values of baseline reproductive rate, relative grandmother years, relative mother years and expected reproductive overlap.

### Analysis

**Live-long versus stop-early.** To understand if species with menopause live or reproduce longer than expected given their size and phylogenetic position we use a model of the form:

$$\tau Z \sim \text{MultiNormal}(\mu, K)$$
$$\tau Z \sim \text{Normal}(\mu Z, \sigma Z)$$
$$\tau S \sim \text{Normal}(\mu S, \sigma S) \quad (3)$$
$$\mu_i = \alpha + \beta_{\text{SIZE}} \tau S_i + \beta_{\text{PR}} M_i$$
$$K_{i,j} = \eta^2 e^{-\rho^2 D_{i,j}}$$

where true lifespan $\tau Z$ is drawn from a multinormal distribution with a mean described by a linear model with terms for size ($\beta_{\text{SIZE}}$) and the effect of menopause ($\beta_{\text{PR}}$). True ordinary maximum lifespan $\tau Z$ and log true size $\tau S$ are drawn from normal distributions described by the means ($\mu Z, \mu S$) and standard deviations ($\sigma Z, \sigma S$) from the lifespan modelling and log observed size, respectively. In species in which menopause is present $M = 1$, otherwise $M = 0$. The dispersion of the multinormal distribution, $K$, is a covariance matrix derived from the phylogenetic distance matrix by the Ornestain–Uhlenbeck process (parameters $\eta, \rho$)[80]. All parameters have weakly informative priors. We use 1,000 time-calibrated bootstrapped phylogenetic trees derived from a recently published cetacean phylogeny[35]. We run the model on each of the bootstrapped phylogenies and combine the posterior estimates to get a complete posterior, which is reported. We use the same modelling structure replacing ordinary maximum lifespan $Z$ with the age at which ovarian activity reaches 0 to understand the relationships between reproductive lifespan, size, phylogeny and the presence or absence of menopause. We confirmed the absence of an effect of menopause on reproductive lifespan by comparing the predictive power of models with and without a menopause parameter (Supplementary Table 5). There is no evidence that a model with a menopause parameter has greater predictive power than a model without a menopause parameter (elpd difference without − with = −0.8 ± 1.2 (±s.e.); Supplementary Table 6).

As the presence of menopause in beluga whales and narwhals[58] and in false killer whales[9] has sometimes proved controversial, we repeat this and other key analyses presented in this study excluding these species. We also repeat the analyses excluding the largest and smallest whale species to confirm that they are not exerting undue influence over our interpretation. Lastly, we also repeated our analysis, coding menopause as a continuous value with error to capture potential differences between menopause species and uncertainties around menopause status. In all cases, these exclusions made no qualitative and little quantitative difference to any of our results (Supplementary Table 5).

**Help and extended lifespans.** We tested three predictions to investigate the role of intergenerational help in the evolution of extended lifespans (Table 1). We compared the grandmother overlap of species with and without menopause using a phylogenetically controlled Bayesian regression model (Table 1). In this model, relative grandmother overlap−calculated as grandmother overlap/age at maturity−is the response variable providing a more meaningful interspecies comparison. Posterior predictive checks demonstrate that this model captures the observed distribution of the data (Supplementary Figs. 5 and 6) and, although the sample size means the true distribution is uncertain, there is no strong evidence that the grandmother years metric is bimodal (Supplementary Fig. 6). If further data find strong evidence of

bimodality it could indicate that there are two modes of life in toothed whales that might have implications for understanding the evolution of life history strategies, including menopause. We use the same analytical pathway to compare the mother overlap of species with and without menopause (Table 1). Last, we used the outputs of the demographic simulations to compare grandmother overlap in species with menopause to their non-menopause ancestor. Applying the kinship demography models to the demographic simulation 'ancestral case' allowed us to generate a distribution of potential relative grandmother overlap values for the proposed non-menopausal ancestor of toothed whale species with menopause.

**Help and reproductive lifespans.** We use two metrics to quantify the costs of offspring and to test the prediction that the offspring of species with menopause are more costly than the offspring of species without menopause (Table 1): age at maturity and size at maturity (adult size). Specifically, we test the predictions that (1) species with menopause have a later age at maturity than expected given the size of their mother, which would suggest extended maternal investment, and (2) species with menopause have larger adult size than expected given their age at maturity, which would suggest greater maternal investment during the juvenile phase. We tested these predictions separately for each sex, using a phylogenetically controlled Bayesian regression model. There are other potential costs of raising offspring not captured by these metrics, including the costs of caring for offspring beyond maturity[28], which could be an interesting focus of future research. In addition, we test the explicit mechanism that some models have proposed by which the costly intergenerational help of older females can benefit their younger kin. These models propose that, by ceasing reproduction, older females can invest in increasing the reproductive rate of their daughters[30,33]. We test this by comparing the observed baseline fecundity in species with menopause, derived from the kinship demography models, compared to the predicted baseline reproductive rate of their non-menopausal rate. In a further analysis, we compare the baseline fecundity of toothed whale species with menopause and without menopause (Supplementary Fig. 7).

**Harm and reproductive lifespans.** We tested two predictions to establish the role of intergenerational harm in the evolution of reproductive lifespan (Table 1). First, we used a phylogenetically controlled Bayesian regression model to compare the predicted reproductive overlap of species with and without menopause, in which reproductive overlap was derived from the kinship demography models (Table 1). Posterior predictive checks demonstrate that this model captures the observed distribution of the data (Supplementary Fig. 5) and the metric does not show any clear evidence of bimodality (Supplementary Fig. 6). We confirm our result by comparing the predictive power of models with and without a menopause parameter (Supplementary Table 7). Similarly, neither running the model on the mean reproductive overlap estimates without uncertainty, nor replacing the true uncertainty around the reproductive overlap with the scaled uncertainty for that species from the grandmother years metric, qualitatively change our conclusions. Second, we compare observed reproductive overlap in species with menopause to the reproductive overlap in the slow life history case representing the non-menopausal analogue of the species with menopause (Table 1). The predicted reproductive overlap is derived by applying kinship demography models to the output of the demographic simulations.

**Female lifespans and male longevity.** We tested two predictions of the male-driven menopause hypothesis. First, we compare relative male and female lifespans in species with and without menopause (Table 1). In species for which estimates of ordinary maximum lifespan are available for both sexes ($n = 30$, including all five species with menopause) we compared the difference in female:male lifespan ratio

$(\tau l)$ in species with ($M_i = 1$) and without ($M_i = 0$) menopause using a model of the form:

$$\delta l \sim \text{logNormal}(\mu, \sigma)$$
$$\mu_i = \alpha + \beta_{\text{PR}} M_i$$
$$\sigma \sim \text{Exp}(1)$$
$$\tau l_{f_i} \sim \text{Normal}(\mu l_{f_i}, \sigma l_{f_i}) \quad (4)$$
$$\tau l_{m_i} \sim \text{Normal}(\mu l_{m_i}, \sigma l_{m_i})$$
$$\delta l_i = \frac{\tau l_{f_i}}{\tau l_{m_i}}$$

where $\tau l_f$ and $\tau l_m$ are, respectively, true female and male lifespans, drawn from a normal distribution parametrized by the distribution of age $Z$ calculated from the posterior of mortality parameters (Lifespan data and modelling section).

Second, for each of the species with menopause we also calculated the probability of each sex reaching the age of female menopause ($l_M$) and the expected lifespan at the age of female menopause ($e_M$). We use the age of last known reproduction as age of menopause to allow all killer whales to be included in the analysis (Kinship demography analysis section) but results are qualitatively similar if the reproductive lifespan is used. We calculated $l_M$ and $e_M$ from the mortality parameters derived from the fitted Gompertz models[82]. Morality parameters for both sexes are calculated in the same model so for each species female and male $l_M$ and $e_M$ can be directly compared from each posterior draw.

### Reporting summary

Further information on research design is available in the Nature Portfolio Reporting Summary linked to this article.

### Data availability

All data used in this analysis are available as a database at: github.com/samellisq/marinelifehistdata.

### Code availability

All R and stan code used for this analysis are available at osf.io/26s7m/. In addition, the mortality model is implemented as an R package available from: github.com/samellisq/marinesurvival.

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

**Acknowledgements** This project was funded as part of a Leverhulme Trust Early Career Research Fellowship awarded to S.E. D.P.C., D.W.F. and M.N.W. acknowledge funding from a NERC Standard Grant (no. NE S010327/1) and M.L.K.N. acknowledges funding from a NERC PhD studentship. We also thank members of the Centre for Research in Animal Behaviour at the University of Exeter for comments throughout the development of this project.

**Author contributions** The project was conceived and designed by S.E. with D.P.C. and D.W.F. Data were gathered and collated by S.E. S.E. designed and implemented the analysis with input from M.L.K.N., M.N.W. and D.W.F. S.E. wrote the first draft of the manuscript with input from D.P.C. All authors contributed to later drafts of the manuscript.

**Competing interests** The authors declare no competing interests.

**Additional information**
**Correspondence and requests for materials** should be addressed to Samuel Ellis.

# Reporting Summary

## Statistics

For all statistical analyses, confirm that the following items are present in the figure legend, table legend, main text, or Methods section.

| n/a | Confirmed | |
|---|---|---|
| ☐ | ☒ | The exact sample size (*n*) for each experimental group/condition, given as a discrete number and unit of measurement |
| ☐ | ☒ | A statement on whether measurements were taken from distinct samples or whether the same sample was measured repeatedly |
| ☒ | ☐ | The statistical test(s) used AND whether they are one- or two-sided<br>*Only common tests should be described solely by name; describe more complex techniques in the Methods section.* |
| ☐ | ☒ | A description of all covariates tested |
| ☐ | ☒ | A description of any assumptions or corrections, such as tests of normality and adjustment for multiple comparisons |
| ☐ | ☒ | A full description of the statistical parameters including central tendency (e.g. means) or other basic estimates (e.g. regression coefficient) AND variation (e.g. standard deviation) or associated estimates of uncertainty (e.g. confidence intervals) |
| ☒ | ☐ | For null hypothesis testing, the test statistic (e.g. *F*, *t*, *r*) with confidence intervals, effect sizes, degrees of freedom and *P* value noted<br>*Give P values as exact values whenever suitable.* |
| ☐ | ☒ | For Bayesian analysis, information on the choice of priors and Markov chain Monte Carlo settings |
| ☐ | ☒ | For hierarchical and complex designs, identification of the appropriate level for tests and full reporting of outcomes |
| ☒ | ☐ | Estimates of effect sizes (e.g. Cohen's *d*, Pearson's *r*), indicating how they were calculated |

*Our web collection on statistics for biologists contains articles on many of the points above.*

## Software and code

Policy information about availability of computer code

| Data collection | Rohatgi, A. WebPlotDigitizer. (2020). Is an online tool used to extract data from figures. |
|---|---|
| Data analysis | R software. Version 4.05. (2021)<br>R package: Tidyverse. Version 1.3.0. (2019).<br>R package: ape. Version 5.0 (2019)<br>R package: marinelifehistory. Version 0.1.0 (2023).<br>R package: marinesurvival. Version 0.1.0. (2023)<br>R package: rstan. Version 2.26.22 (2023)<br>R package: cmstanr. Version 0.5.3 (2023)<br>Code and data from this study: github.com/samellisq/marinelifehistdata, github.com/samellisq/marinesurvival, osf.io/26s7m/ . |

For manuscripts utilizing custom algorithms or software that are central to the research but not yet described in published literature, software must be made available to editors and reviewers. We strongly encourage code deposition in a community repository (e.g. GitHub). See the Nature Portfolio guidelines for submitting code & software for further information.

## Data

Policy information about availability of data

All manuscripts must include a data availability statement. This statement should provide the following information, where applicable:
- Accession codes, unique identifiers, or web links for publicly available datasets
- A description of any restrictions on data availability
- For clinical datasets or third party data, please ensure that the statement adheres to our policy

All data used in this analysis are avalaible as a database at: github.com/samellisq/marinelifehistdata.

## Research involving human participants, their data, or biological material

Policy information about studies with human participants or human data. See also policy information about sex, gender (identity/presentation), and sexual orientation and race, ethnicity and racism.

| | |
|---|---|
| Reporting on sex and gender | NA |
| Reporting on race, ethnicity, or other socially relevant groupings | NA |
| Population characteristics | NA |
| Recruitment | NA |
| Ethics oversight | NA |

Note that full information on the approval of the study protocol must also be provided in the manuscript.

# Field-specific reporting

Please select the one below that is the best fit for your research. If you are not sure, read the appropriate sections before making your selection.

☐ Life sciences   ☐ Behavioural & social sciences   ☒ Ecological, evolutionary & environmental sciences

For a reference copy of the document with all sections, see nature.com/documents/nr-reporting-summary-flat.pdf

# Ecological, evolutionary & environmental sciences study design

All studies must disclose on these points even when the disclosure is negative.

| | |
|---|---|
| Study description | This study uses published age-structured toothed whale life-history data to understand the evolution of menopause in toothed whales. |
| Research sample | To our knowledge, the sample consists of all published age-structured toothed whale datasets. |
| Sampling strategy | Sample size was maximised over as many species as possible. Overall it consisted of 269 datasets allowing the calculation of lifespan for 32 species of female and 33 species of male toothed whale. |
| Data collection | Data were identified in publications through systematic and opportunistic literature searches. Data were then extracted using webplotdigitiser if in a figure or by transcription if in a table. |
| Timing and spatial scale | Literature search conducted during  May-September 2021 |
| Data exclusions | A given species-sex were only used for further analysis if they had at least one dataset with more than 10 samples and a sampling rate (n samples/max observed age) of greater than 0.5/ |
| Reproducibility | There was no experiment in this study. |
| Randomization | There was no experiment in this study. |
| Blinding | Blinding was not possible because all data are from species of known identity. |

Did the study involve field work?   ☐ Yes   ☒ No

# Reporting for specific materials, systems and methods

We require information from authors about some types of materials, experimental systems and methods used in many studies. Here, indicate whether each material, system or method listed is relevant to your study. If you are not sure if a list item applies to your research, read the appropriate section before selecting a response.

## Materials & experimental systems

| n/a | Involved in the study |
|-----|----------------------|
| ☒ ☐ | Antibodies |
| ☒ ☐ | Eukaryotic cell lines |
| ☒ ☐ | Palaeontology and archaeology |
| ☒ ☐ | Animals and other organisms |
| ☒ ☐ | Clinical data |
| ☒ ☐ | Dual use research of concern |
| ☒ ☐ | Plants |

## Methods

| n/a | Involved in the study |
|-----|----------------------|
| ☒ ☐ | ChIP-seq |
| ☒ ☐ | Flow cytometry |
| ☒ ☐ | MRI-based neuroimaging |

## Plants

| | |
|--|--|
| Seed stocks | NA |
| Novel plant genotypes | NA |
| Authentication | NA |

