## [Peer Review File · Nature]

Manuscript Title: The evolution of menopause in toothed whales

Reviewer Comments & Author Rebuttals

Reviewer Reports on the Initial Version:

Referees' comments:

Referee #1 (Remarks to the Author):

This paper uses data from multiple species of toothed whales to test 5 hypotheses about the evolution of menopause, making use of the observation that menopause has evolved multiple times in this taxon. This is a nice approach, and the authors have gone to impressive lengths to extract data from the literature to reconstruct demographic parameters which can be used to test these hypotheses in different whale species. Overall, I think this paper is a very useful contribution to the literature on the evolution of menopause, and likely to be of broad interest. Much of the paper is clearly written, but I do have some comments about the lack of clarity surrounding one of their hypothesis tests, and about the data they use to test all of their hypotheses.

The first two hypotheses relate to whether menopause evolves because overall lifespan (but not reproductive lifespan) was extended, leading to long post-reproductive life, or whether menopause evolved by shortening the reproductive lifespan. The authors find that species with menopause have longer overall lifespans, but not reproductive lifespans, than species without menopause, suggesting menopause evolved by lengthening overall lifespan, rather than shortening reproductive lifespan. This section is clear and the authors appear to have used appropriate analyses to test their hypothesis.

They then test a further three hypotheses: hypotheses 3 and 4 are related to whether extended post-reproductive lifespans evolved because of the provision of intergenerational help, and the final, fifth, hypothesis relates to whether menopause evolved in order to avoid intergenerational conflict (caused by two generations reproducing simultaneously).

The first and last of these hypotheses are clearly explained and seem reasonable – the authors test for the potential for intergenerational help by examining whether there is greater generational overlap in species with menopause, compared to species without menopause (the authors find there is the potential for help). And the fifth hypothesis finds that menopause is associated with greater generational, but not reproductive overlap, suggesting avoidance of reproductive conflict may be important in the evolution of menopause.

I have more concerns with hypothesis 4, which tests whether older females increase the fitness of younger generations. The authors conclude there is no evidence that older females do increase the fitness of younger generations, based on tests of reproductive rate and the costs of raising offspring. I

find this section quite confusing. Firstly, the results of hypotheses 3 and 4 appear to contradict one another – results of hypothesis 3 suggests there is potential for intergenerational help (and that this can explain extended lifespan) but results for hypothesis 4 suggests that older females don't increase the fitness of younger generations, implying there is no intergenerational help provided (or at least, help isn't sufficient to impact fitness). In the Discussion, the authors mention other ways that older females might provide help to younger generations (which are not empirically tested here, presumably because sufficient data is not available). Do they think these alternative forms of help are sufficient to increase the fitness of younger relatives? Because if older females aren't increasing the fitness of younger generations, then it isn't clear to me why an extended lifespan would evolve? I understand the authors find support for their final hypothesis that reproductive lifespan is not extended because of the costs of intergenerational conflict, but this doesn't explain why overall lifespan is extended. I think the authors need to clarify this section – do they mean that the empirical tests they present here of whether older females increase the fitness of younger generations are not supported, but that tests using other kinds of data might support this hypothesis? If so, this could be clearly stated in the paper.

Secondly, I have some concerns with the analyses used to test hypotheses 4:

1. The first analysis – of whether menopause is associated with higher reproductive rate – tests whether species with menopause have higher reproductive rates than their hypothetical non-menopause ancestor (if I understand correctly, the hypothetical non-menopausal ancestor is assumed to have the lifespan expected of a whale species of the same size i.e. a shorter lifespan than observed in the species with menopause, but the same reproductive lifespan as observed in that species with menopause). But the other hypothesis tests all involve comparing species with menopause to those without menopause on whatever the outcome of interest is, i.e. using species without menopause as the 'control' against which species with menopause are compared. Sometimes these analyses are supported with additional analysis which compares species with menopause against their hypothetical non-menopausal ancestor. However, the test of reproductive rate is the only test which does not involve comparing species with and without menopause – why was the comparison of species with and without menopause not performed here?

2. According to figure 3 (which relates to the analyses performed for this hypothesis test), for the analysis of reproductive rate, species with menopause was also compared against another hypothetical species – one with the same lifespan as the contemporary species with menopause but which continues to reproduce until the end of life (the authors refer to this hypothetical species as the "slow life history" case). This analysis is not mentioned in the main text, and – although the methods for this analysis are described in the methods section – no clear justification is given for this additional analysis. It therefore isn't obvious why this analysis is included in the figure, but not discussed in the main text. It might help the reader navigate the several hypotheses tested here if the methods were structured so that a clear explanation, including justification, was given for each hypothesis test. Currently, the methods are structured by type of methods (e.g. kinship demography analysis, demographic simulations) rather than explicitly by hypothesis test, which makes it a little hard to work out exactly what was done for each hypothesis test, and why.

3. The authors also include an analysis of whether offspring are more costly to raise in species with menopause when testing hypothesis 4. They do this by comparing age and size at maturity in species

with and without menopause. While this seems reasonable, it also doesn't seem to cover all the possibilities of higher costs of raising offspring in species with menopause (especially given that investment seems to continue after maturity in at least some species). I'd suggest the authors perhaps include some discussion about whether their tests of the costs of raising offspring cover all possibilities or not.

I also have some comments concerning the data used to test hypotheses. Data on age patterns of mortality and fertility are largely gleaned from whales who died during mass stranding events. Obviously data on whale demography are challenging to come by (certainly compared to demographic data on our own species), and the authors appear to have thoroughly searched the existing literature to extract all available data on this topic. But there will be limitations with this data. For example, if I understand correctly, the authors reconstruct age patterns of mortality and fertility for a species provided more than 10 samples over the age of maturity were observed for that species. This seems a rather small sample size to base the demographic parameters of an entire species on. Was the demography of many species reconstructed from such small samples? It would be useful to include a table with summary statistics on the raw data used to reconstruct demographic patterns. The authors make available their data online, which is great, but it would still be helpful for the reader to easily see sample sizes for all species included in the supplementary materials. Were any sensitivity analyses run to exclude species with particularly small sample sizes, to determine whether results might be biased by the inclusion of species with large uncertainty around demographic parameters. And how representative are samples of whales who die in stranding events? are there any biases in the whales who are stranded? It would be good to include some additional discussion about how data limitations might potentially affect results.

My final comment is minor, and relates to the authors' description of the literature on human menopause. They make some quite definitive statements about human menopause, such as that "research has convincingly demonstrated that the presence of mothers and grandmothers increases the fitness of their offspring and grandoffspring" (lines 43-46). While I consider this statement to be accurate for mothers, I'm not quite so sure about grandmothers. The statement is certainly plausible, given that grandmothers do invest significantly in their grandchildren. And there are often associations found between grandmother presence and/or investment and improved child outcomes, but these associations are not always seen, and there are still some debates about whether grandmothers causally improve their grandchildren's fitness (e.g. Sadruddin et al [2019] How do grandparents influence child health and development? A systematic review, *Social Science & Medicine*, Volume 239). So I might be inclined to make a slightly less definitive statement on whether grandmothers do increase their grandchildren's fitness

Referee #2 (Remarks to the Author):

The authors present an authoritative, comparative, comprehensive and highly quantitative study to address a long-standing question in evolutionary biology: which traits is selection acting upon that result

in the evolution of menopause.

The authors build on their own impressive body of work, which provide the foundations on which to base this cross-species comparative study.

The study system of toothed whales arguably offers a more suitable model with which to test alternative hypotheses than primates, as there are multiple species within the odontocetes for which menopause has been identified, but with limited phylogenetic congruence. The outcomes of the study are of course informative of our own evolutionary history, and the factors that have driven the evolution of a long post-reproductive female life span in humans. As such, this article will be of broad interest to both the scientific community and the general public alike. It is therefore highly suited for publication in Nature.

Step-by step the authors show that the menopause in tooth whales is due to extended lifespan rather than shortened reproductive lifespan; and that intergenerational help explains selection on lengthening life span, while prevention of inter-generational conflict constrains reproductive lifespan.

One of the most surprising things to me, is that given the benefits of intergenerational care, why we this trait is not more common and why it differs between even the most closely related of species that have similar kin-based social structure such as the long- and short-finned pilot whales. The authors address this outstanding question in the paragraph lines 294-316. The efficacy of selection on late life traits is also dependent on the size on the proportion of the population reaching that life stage. Differences in mortality doubling rate between long- and short-finned pilot whales have been proposed to explain why selection on such late-life traits may vary between taxa. This could be noted in this paragraph.

The writing and presentation is largely very clear, so I have just a few minor comments:

The introduction is quite long for a journal like Nature. There is a little redundancy in the text that could be cut to make the opening paragraphs more succinct. For example, lines 47-52 and later lines 74-79 repeat much of the same information.

Figure 1 legend, line 117. As panel C is modelling support for the stop-early hypothesis, I think it would make more sense to highlight that "Species with menopause (red) do not have shorter reproductive lifespans than expected given their size." as opposed to "longer" as in the current text.

Also Figure 1, panel C. It looks like the killer whale and one non-menopause species >5metres are missing?.

Could a version of panels B & C with labels for the non-menopausal species be added to the supplementary materials.

The two largest species look to potentially have a large effect on the model fit. I guess these are the sperm whale and Northern bottlenose whale? I know the authors controlled for phylogenetic relatedness, but I think it would be helpful for the reader and future studies to understand which

species may have influenced the pattern.

Referee #3 (Remarks to the Author):

This is an exciting study that tests hypotheses about the evolution of menopause using a comparative approach in toothed whales, in which menopause has evolved independently in at least 4 distinct lineages. The main result of the study is that toothed whale species with menopause live longer than expected but have reproductive lifespans that are typical, given their body size, supporting the “live-long hypothesis” for the evolution of menopause. This result is well-supported, and its presentation in Figure 1b and 1c is particularly striking.

There are three sets of supporting results that evaluate some of the factors that have been proposed to influence the evolution of menopause, including the benefits of intergenerational help and the costs of intergenerational competition.

The first set of supporting results finds evidence for the idea that, by living longer, species with menopause benefit from intergenerational help. This includes three separate sub-results:

- Species with menopause have greater relative overlap between the lifespans of grandmothers and grandoffspring than species without menopause.
- The same pattern was found for the degree of relative overlap between the lifespans of mothers and offspring, but the effect size was smaller.
- Sexual dimorphism in body size is greater in species with menopause, suggesting greater allocation of maternal or grandmaternal care to male offspring.

These first two sub-results establish that species with menopause have greater opportunities for intergenerational help because grandmother/grandoffspring and mother/offspring pairs have greater overlap in their lifespan compared to extant species without menopause or to their hypothetical non-menopausal ancestors. In some sense, these results feel inevitable given the main result that these species have extended lifespans. It would be powerful if species differences in social behavior and social organization could be brought to bear on the question of whether intergenerational help actually occurs more often in these species, but I recognize that the data probably aren't available. The sub-result about sexual dimorphism in body size was the least convincing to me because this is so heavily influenced by competitive regimes, and the effect appeared weak and possibly influenced by two outliers (vaquita and baiji). It also does not find a parallel in the evolution of menopause in humans, which have considerably less sexual dimorphism in body size compared to their hominid relatives orangutans and gorillas, and comparable sexual dimorphism compared to chimpanzees, all which of course do not have menopause as humans do. Both the hypothesis and result about sexual dimorphism in body size do not seem very robust, given the small sample size and lack of accounting for social factors.

The second set of supporting results tests whether the greater opportunities for intergenerational help in species with menopause are associated with either increased female reproductive rate relative to an

ancestor without menopause, or more costly offspring relative to species without menopause. Using demographic simulations, the authors find that reproductive rates are slower in the species with menopause compared to the ancestral condition. They also find that offspring in species with menopause were not more costly based on proxy measures (age at maturity and adult body size) that did not differ among toothed whale species depending on the presence of menopause.

The third set of supporting results finds that by extending the lifespan without also extending the reproductive lifespan, species with menopause avoid potential costs in the form of intergenerational competition for resources. Specifically, (1) mother-daughter reproductive overlap did not differ according to the presence of menopause; and (2) in species with menopause, simulations indicate that mother-daughter reproductive overlap would be greater if females were to continue reproducing. While the patterns described are persuasive and robust, these findings also seem foreseeable given the finding that reproductive lifespans relative to body size remain typical in the species with menopause.

Originality and significance

The authors do a fine job of framing the study's significance by drawing parallels to the evolution of menopause in humans, but I have a few minor quibbles. The take-home statement of significance occurs on lines 318–319: “our results show that the same pathway that lead to the evolution of menopause in humans also lead to the evolution of menopause in toothed whales” Note that both instances of “lead” in this sentence should instead be “led,” the past tense of “lead” (the similar sentence in the abstract on line 27 correctly uses “led”). I found this framing about “the same pathway” a bit odd—as though there is a single, unambiguous, widely accepted set of factors that explains the evolution of menopause in humans. I suppose the authors are using “pathway” to refer to finding strong support for “live long” as opposed to “stop early” in both humans and toothed whales. But elsewhere in the manuscript (line 40), the authors state that “the pathways and mechanisms by which menopause evolves remain debated,” and the most recent review article cited in this section concludes with “The question of why prolonged life after the cessation of fertility has evolved in some species has not been fully answered” (Croft et al. 2014, p. 414). Thus, the assertion that the evolution of menopause in humans and toothed whales occurred “by the same pathway” may be a bit too sweeping.

Data & methodology: validity of approach, quality of data, quality of presentation

I had no major concerns about the data, methodology, or the validity of the approach. As someone who does not work with toothed whales, I am not familiar with the age determination method based on teeth or the reproductive lifespan determination method based on ovarian corpora. But they appear to be well validated.

The quality of the presentation was sufficient (Figure 1 was particularly nice). In Figure 2, I think it would help to use a bit more visual contrast between the line ranges representing the 50% and 95% credible intervals (e.g., by making the 50% intervals slightly thicker or darker).

I had many issues with the supplementary figures.

- Fig. S1: The labels with species common names are cut off. The filled horizontal bars are redundant, as you could just show the points and line ranges with the same color coding. The salmon/teal color coding for used here to show presence or absence of menopause differs from the red/blue coding of figures in the manuscript. The color coding is not explained in a legend or in the caption. The figure is supposed to show that species with menopause have smaller female/male body size ratios, but the visual presentation is not convincing (the menopause species seem to fall squarely within the range of other species) and suggest that the result may be driven primarily extreme sexual dimorphism in the opposite direction in the vaquita and baiji.
- Fig. S2: Uninformative axis labels (e.g., “log.size” vs. “Size (log)”) with no units and inconsistent capitalization. What do the fitted lines and gray shaded regions in each panel show? Why do these seem to fit the data quite poorly by passing well below most of the points (e.g., in panel a there are 6 reddish points below the line and at least 21 above the line)? The caption says “Females of species with menopause do not produce larger female or male offspring given their size,” but the figure does not seem to show this. Instead, it seems to show offspring age at maturity given the female’s body size. Same issue as noted above with salmon/teal color coding.
- Fig. S3: Many of the same issues as in Fig. S2.
- Fig. S4: I couldn’t find a reference to this figure anywhere in the text. Why is there a gray entry for NA in the legend? What does “(age Z)” mean in the x-axis title? The y-axis and caption are not entirely consistent (is it age at last reproduction [e.g., a mean], or oldest known reproduction?)

There are two documents called Supplementary 2. The supplementary document about phylogeny, size, etc. has a very informal style with many typos—it looks and reads like a rough draft and should be proofread more carefully. There are citations in this document but no bibliography. I will admit that I did not get far into this document with careful reading.

Overall, this was an exciting, high-quality study with a couple of compelling topline results followed by an assortment of less interesting supporting results that seem to confirm what seem like the logical consequences of living longer without changing the reproductive lifespan, or test predictions about intergenerational help and conflict using indirect proxies.

Referee #4 (Remarks to the Author):

Review of: The Evolution of Menopause in Toothed Wales

This paper brings together: (1) an impressive set of data, collated across a variety of sources; (2) cutting edge demographic analysis methods and; (2) sophisticated use of simulation, to make inferences about the most likely causes of the evolution of menopause in toothed whales.

The paper is well executed and the review of different hypotheses is very clear. The combined evidence and simulations come down in favour of the ‘live-long’ hypothesis: it is the post-reproductive lifespan

that was most likely selected for, rather than the 'stopping-early' hypothesis in which reproductive cessation is selected for. The causal argument put forward is that post-reproductive lifespans simultaneously allow older females to help provision and / or support younger generations while avoiding reproductive competition with younger females.

I appreciated the effort to write this paper for a broad audience, though I found myself wanting more methodological details peppered throughout the text rather than having to wait for the methods section to learn about the results in detail. Key definitions, for example the definition of grandmother overlap, could have been provided when first introduced as they are not necessarily as intuitive as the text implies. This seems important if the comparison to humans is to remain a prominent feature of the paper (see comments below). In places, the effect sizes seem relatively small for such long-lived species, e.g. less than one year difference in relative grand offspring years between species with and without menopause (Lines 140-142), so I found myself wanting more help interpreting the importance of the different results relative to each other.

My major comment on the paper is that the inference from these analyses to humans is probably stretched beyond what is justified by either the present analyses or the evidence for human menopause evolution. The paper is framed around the usefulness of the presented data for understanding the human case. Really the whales are interesting enough in their own right (and I say this a person who researches humans!), so I would suggest toning down or editing the arguments about humans. I have two main reasons for this critique.

First, as of course the authors know, we don't have phylogenetically close species comparisons for humans. However, while certain demographic and life-history features that differ across toothed whale species with and without menopause are potentially illuminating and / or consistent with those observed in humans, there are many other unobserved factors that make a straightforward comparative inference about major evolutionary forces of human menopause / post-reproductive lifespan evolution difficult.

Second, important causal mechanisms - like reproductive overlap across generations and indeed, the assumption of natural survival and fertility in humans - are almost certainly culturally regulated. All human populations have (highly variable) rules about, for example, who can marry / have sex with whom, in what place and at what age - and so evidence that there is more or less reproductive overlap in humans will vary by cultural and ecological context (see Mace & Alvergne 2012; Koster et al 2019) - there isn't really a species-wide measure that could serve as a comparator to the toothed whale data. Indeed, menopause progression itself appears to be variable (within constraints) across human populations, progressing differently in populations with different kinship structures (see for example, Snopkowski, Moya & Sear 2014). There is also a substantial debate in the human literature, so claims that the evidence for one or other evolutionary hypothesis for human menopause evolution is "overwhelming" is itself as yet too strong.

For these reasons I don't find it persuasive to argue that the mechanisms driving the evolution of

menopause are even likely the same in toothed whales and humans. Unfortunately we simply don't have the evidentiary basis to make such a comparative claim. That's not to say that human researchers shouldn't consult this paper when thinking about the evolution of menopause / post-reproductive lifespan. They absolutely should.

Mace, R., & Alvergne, A. (2012). Female reproductive competition within families in rural Gambia. *Proceedings of the Royal Society B: Biological Sciences*, 279(1736), 2219-2227.

Koster J. et al. (2019) Kinship ties across the lifespan in human communities." *Philosophical Transactions of the Royal Society B* 374.1780: 20180069.

Snopkowski, K., Moya, C., & Sear, R. (2014). A test of the intergenerational conflict model in Indonesia shows no evidence of earlier menopause in female-dispersing groups. *Proceedings of the Royal Society B: Biological Sciences*, 281(1788), 20140580.

Author Rebuttals to Initial Comments:

We would like to take this opportunity to thank the reviewers for their positive opinion of our manuscript and constructive suggestions. We feel their input has greatly improved the manuscript.

Referee #1 (Remarks to the Author):

This paper uses data from multiple species of toothed whales to test 5 hypotheses about the evolution of menopause, making use of the observation that menopause has evolved multiple times in this taxon. This is a nice approach, and the authors have gone to impressive lengths to extract data from the literature to reconstruct demographic parameters which can be used to test these hypotheses in different whale species. Overall, I think this paper is a very useful contribution to the literature on the evolution of menopause, and likely to be of broad interest. Much of the paper is clearly written, but I do have some comments about the lack of clarity surrounding one of their hypothesis tests, and about the data they use to test all of their hypotheses.

The first two hypotheses relate to whether menopause evolves because overall lifespan (but not reproductive lifespan) was extended, leading to long post-reproductive life, or whether menopause evolved by shortening the reproductive lifespan. The authors find that species with menopause have longer overall lifespans, but not reproductive lifespans, than species without menopause, suggesting menopause evolved by lengthening overall lifespan, rather than shortening reproductive lifespan. This section is clear and the authors appear to have used appropriate analyses to test their hypothesis.

They then test a further three hypotheses: hypotheses 3 and 4 are related to whether extended post-reproductive lifespans evolved because of the provision of intergenerational help, and the final, fifth, hypothesis relates to whether menopause evolved in order to avoid intergenerational conflict (caused by two generations reproducing simultaneously).

The first and last of these hypotheses are clearly explained and seem reasonable – the authors test for the potential for intergenerational help by examining whether there is greater generational overlap in species with menopause, compared to species without menopause (the authors find there is the potential for help). And the fifth hypothesis finds that menopause is associated with greater generational, but not reproductive overlap, suggesting avoidance of reproductive conflict may be important in the evolution of menopause.

I have more concerns with hypothesis 4, which tests whether older females increase the fitness of younger generations. The authors conclude there is no evidence that older females do increase the fitness of younger generations, based on tests of reproductive rate and the costs of raising offspring. I find this section quite confusing. Firstly, the results of hypotheses 3 and 4 appear to contradict one another – results of hypothesis 3 suggests there is potential for intergenerational help (and that this can explain extended lifespan) but results for hypothesis 4 suggests that older females don't increase the fitness of younger generations, implying there is no intergenerational help provided (or at least, help isn't sufficient to impact fitness). In the Discussion, the authors mention other ways that older females

might provide help to younger generations (which are not empirically tested here, presumably because sufficient data is not available). Do they think these alternative forms of help are sufficient to increase the fitness of younger relatives? Because if older females aren't increasing the fitness of younger generations, then it isn't clear to me why an extended lifespan would evolve? I understand the authors find support for their final hypothesis that reproductive lifespan is not extended because of the costs of intergenerational conflict, but this doesn't explain why overall lifespan is extended. I think the authors need to clarify this section – do they mean that the empirical tests they present here of whether older females increase the fitness of younger generations are not supported, but that tests using other kinds of data might support this hypothesis? If so, this could be clearly stated in the paper.

We thank the reviewer for raising this important point and we apologise that the aim and scope of this analysis was not clear in the original manuscript. Although this analysis finds no effect of the presence of menopause on female reproductive rate there are a variety of other fitness benefits that can and do occur, including increasing the survival of grandoffspring and offspring (e.g. Foster et al 2012 [ref. list 21], Natrass et al 2019[20]; lines 294-300). We had designed this fecundity analysis as a specific test of a mechanism of intergenerational help suggested by previous theoretical work to be central to the evolution of menopause (Chan et al 2016, 2017: [29, 32]). This was not intended to be a general test of the fitness benefits of menopause, but it clearly has implications for this. In killer whales we have shown previously that the major benefits that post-reproductive females provide their kin is directed to their sons rather than their daughters (Foster et al 2012 [21], Weiss et al 2023 [27]) and thus it is perhaps unsurprising that we don't find an effect of post-reproductive lifespan on female fecundity. We have now re-written the 'Intergenerational help and reproductive lifespan' section (lines 172-211) to clarify the aims and limits of this analysis. As suggested by the reviewer, we have also added a discussion linking this result to the fitness benefits of menopause in humans and toothed whales.

Secondly, I have some concerns with the analyses used to test hypotheses 4:

1. The first analysis – of whether menopause is associated with higher reproductive rate – tests whether species with menopause have higher reproductive rates than their hypothetical non-menopause ancestor (if I understand correctly, the hypothetical non-menopausal ancestor is assumed to have the lifespan expected of a whale species of the same size i.e. a shorter lifespan than observed in the species with menopause, but the same reproductive lifespan as observed in that species with menopause). But the other hypothesis tests all involve comparing species with menopause to those without menopause on whatever the outcome of interest is, i.e. using species without menopause as the 'control' against which species with menopause are compared. Sometimes these analyses are supported with additional analysis which compares species with menopause against their hypothetical non-menopausal ancestor. However, the test of reproductive rate is the only test which does not involve comparing species with and without menopause – why was the comparison of species with and without menopause not performed here?

We did not include a general comparison because this analysis is designed to test a specific model prediction. As outlined above, we have rewritten this section (lines 172-211) to make this clearer. Further, we also now include an interspecies comparison of baseline reproductive rate in the supplementary

material (figure S5). Consistent with our originally reported result, we find that species with menopause have the same rate of reproduction as expected given their size and phylogenetic position.

2. According to figure 3 (which relates to the analyses performed for this hypothesis test), for the analysis of reproductive rate, species with menopause was also compared against another hypothetical species – one with the same lifespan as the contemporary species with menopause but which continues to reproduce until the end of life (the authors refer to this hypothetical species as the “slow life history” case). This analysis is not mentioned in the main text, and – although the methods for this analysis are described in the methods section – no clear justification is given for this additional analysis. It therefore isn’t obvious why this analysis is included in the figure, but not discussed in the main text. It might help the reader navigate the several hypotheses tested here if the methods were structured so that a clear explanation, including justification, was given for each hypothesis test. Currently, the methods are structured by type of methods (e.g. kinship demography analysis, demographic simulations) rather than explicitly by hypothesis test, which makes it a little hard to work out exactly what was done for each hypothesis test, and why.

As suggested by the reviewer, we have now expanded our methods section to lay out the analyses performed to test each hypothesis (lines 567-723 table 1).

We also apologise for causing confusion- in our previous draft, we referred to figure 3 as a general ‘reference’ for our demographic simulation methods. But as the reviewer highlights, this figure does not represent any results described in this section. This reference to figure 3 has now been removed.

We have also included a description of the slow life history case in the main text (lines 235-238).

3. The authors also include an analysis of whether offspring are more costly to raise in species with menopause when testing hypothesis 4. They do this by comparing age and size at maturity in species with and without menopause. While this seems reasonable, it also doesn’t seem to cover all the possibilities of higher costs of raising offspring in species with menopause (especially given that investment seems to continue after maturity in at least some species). I’d suggest the authors perhaps include some discussion about whether their tests of the costs of raising offspring cover all possibilities or not.

This discussion has been included as suggested (lines 203-211).

I also have some comments concerning the data used to test hypotheses. Data on age patterns of mortality and fertility are largely gleaned from whales who died during mass stranding events. Obviously data on whale demography are challenging to come by (certainly compared to demographic data on our own species), and the authors appear to have thoroughly searched the existing literature to extract all available data on this topic. But there will be limitations with this data. For example, if I understand correctly, the authors reconstruct age patterns of mortality and fertility for a species provided more than 10 samples over the age of maturity were observed for that species. This seems a rather small

sample size to base the demographic parameters of an entire species on. Was the demography of many species reconstructed from such small samples? It would be useful to include a table with summary statistics on the raw data used to reconstruct demographic patterns. The authors make available their data online, which is great, but it would still be helpful for the reader to easily see sample sizes for all species included in the supplementary materials. Were any sensitivity analyses run to exclude species with particularly small sample sizes, to determine whether results might be biased by the inclusion of species with large uncertainty around demographic parameters. And how representative are samples of whales who die in stranding events? are there any biases in the whales who are stranded? It would be good to include some additional discussion about how data limitations might potentially affect results.

As per the reviewer's useful suggestion we now include a table of sample sizes for the lifespan (table S1) and fertility (table S2) data. For age-specific mortality, species were included if they had at least one dataset with >10 samples and a sampling rate (number of samples / oldest observed whale for that species-sex) (lines 506-508). The median number of whales for species included in the dataset was 98, and the lowest quartile is 43.5 (table S1).

Our analytical pathway is designed to accommodate a range of sample sizes and our results are therefore robust to the variation in sample sizes in our data. In particular, under small sample sizes the uncertainty in the model is high, resulting in wide credible intervals around the estimates of maximum lifespan. We carry this uncertainty downstream through all subsequent analysis by always modelling uncertainty in our lifespan estimates in statistical models. In parallel to the work presented in this manuscript, we have undertaken extensive simulations which demonstrate that even under small sample sizes, this uncertainty means that the true lifespan is captured within the credible interval of the predicted lifespan (<https://doi.org/10.1101/2023.02.22.529527>; referenced in text lines 528-230).

Finally, as suggested, we now include a discussion of the data sources lines 511-514.

My final comment is minor, and relates to the authors' description of the literature on human menopause. They make some quite definitive statements about human menopause, such as that "research has convincingly demonstrated that the presence of mothers and grandmothers increases the fitness of their offspring and grandoffspring" (lines 43-46). While I consider this statement to be accurate for mothers, I'm not quite so sure about grandmothers. The statement is certainly plausible, given that grandmothers do invest significantly in their grandchildren. And there are often associations found between grandmother presence and/or investment and improved child outcomes, but these associations are not always seen, and there are still some debates about whether grandmothers causally improve their grandchildren's fitness (e.g. Sadruddin et al [2019] How do grandparents influence child health and development? A systematic review, *Social Science & Medicine*, Volume 239). So I might be inclined to make a slightly less definitive statement on whether grandmothers do increase their grandchildren's fitness

In response to this comment, in the revised manuscript we have clarified our discussion of the literature human menopause as suggested by the reviewer (lines 41-47).

Referee #2 (Remarks to the Author):

The authors present an authoritative, comparative, comprehensive and highly quantitative study to address a long-standing question in evolutionary biology: which traits is selection acting upon that result in the evolution of menopause.

The authors build on their own impressive body of work, which provide the foundations on which to base this cross-species comparative study.

The study system of toothed whales arguably offers a more suitable model with which to test alternative hypotheses than primates, as there are multiple species within the odontocetes for which menopause has been identified, but with limited phylogenetic congruence. The outcomes of the study are of course informative of our own evolutionary history, and the factors that have driven the evolution of a long post-reproductive female life span in humans. As such, this article will be of broad interest to both the scientific community and the general public alike. It is therefore highly suited for publication in Nature.

Step-by step the authors show that the menopause in tooth whales is due to extended lifespan rather than shortened reproductive lifespan; and that intergenerational help explains selection on lengthening life span, while prevention of inter-generational conflict constrains reproductive lifespan.

One of the most surprising things to me, is that given the benefits of intergenerational care, why we this trait is not more common and why it differs between even the most closely related of species that have similar kin-based social structure such as the long- and short-finned pilot whales. The authors address this outstanding question in the paragraph lines 294-316. The efficacy of selection on late life traits is also dependent on the size on the proportion of the population reaching that life stage. Differences in mortality doubling rate between long- and short-finned pilot whales have been proposed to explain why selection on such late-life traits may vary between taxa. This could be noted in this paragraph.

We thank the reviewer for this suggestion and we have included a discussion of mortality rate-doubling time in the discussion (lines 314-339).

The writing and presentation is largely very clear, so I have just a few minor comments:

The introduction is quite long for a journal like Nature. There is a little redundancy in the text that could be cut to make the opening paragraphs more succinct. For example, lines 47-52 and later lines 74-79 repeat much of the same information.

We have now shortened the introduction as suggested by the reviewer (lines 23-63). Specifically- and again as suggested by the reviewer- we have moved the discussion of the live-long and stop-early hypothesis to the first paragraph of the live-long stop-early section, consolidating the introduction of the hypotheses with their predictions (lines 65-76). We thank the reviewer for this suggestion and feel it has increased the clarity of the manuscript.

Figure 1 legend, line 117. As panel C is modelling support for the stop-early hypothesis, I think it would make more sense to highlight that "Species with menopause (red) do not have shorter reproductive lifespans than expected given their size." as opposed to "longer" as in the current text.

Change made as suggested (lines 115-117).

Also Figure 1, panel C. It looks like the killer whale and one non-menopause species >5metres are missing?.

Only species where reproductive data are available can be included in the plot/analysis represented in figure 1c. We have edited the figure legend to make this clear (lines 114-115).

Could a version of panels B & C with labels for the non-menopausal species be added to the supplementary materials.

This figure has been added as suggested. Figure S1.

The two largest species look to potentially have a large effect on the model fit. I guess these are the sperm whale and Northern bottlenose whale? I know the authors controlled for phylogenetic relatedness, but I think it would be helpful for the reader and future studies to understand which species may have influenced the pattern.

We thank the reviewer for raising this important point. The largest two species in both figures 1b and c are sperm whales and Baird's beaked whales. Excluding these species (and/or northern bottlenose whales) from the analysis makes no quantitative and little qualitative differences to the results (table S5).

Referee #3 (Remarks to the Author):

This is an exciting study that tests hypotheses about the evolution of menopause using a comparative approach in toothed whales, in which menopause has evolved independently in at least 4 distinct lineages. The main result of the study is that toothed whale species with menopause live longer than expected but have reproductive lifespans that are typical, given their body size, supporting the “live-long hypothesis” for the evolution of menopause. This result is well-supported, and its presentation in Figure 1b and 1c is particularly striking.

There are three sets of supporting results that evaluate some of the factors that have been proposed to influence the evolution of menopause, including the benefits of intergenerational help and the costs of intergenerational competition.

The first set of supporting results finds evidence for the idea that, by living longer, species with menopause benefit from intergenerational help. This includes three separate sub-results:

- Species with menopause have greater relative overlap between the lifespans of grandmothers and grandoffspring than species without menopause.
- The same pattern was found for the degree of relative overlap between the lifespans of mothers and offspring, but the effect size was smaller.
- Sexual dimorphism in body size is greater in species with menopause, suggesting greater allocation of maternal or grandmaternal care to male offspring.

These first two sub-results establish that species with menopause have greater opportunities for intergenerational help because grandmother/grandoffspring and mother/offspring pairs have greater overlap in their lifespan compared to extant species without menopause or to their hypothetical non-menopausal ancestors. In some sense, these results feel inevitable given the main result that these species have extended lifespans. It would be powerful if species differences in social behavior and social organization could be brought to bear on the question of whether intergenerational help actually occurs more often in these species, but I recognize that the data probably aren't available. The sub-result about sexual dimorphism in body size was the least convincing to me because this is so heavily influenced by competitive regimes, and the effect appeared weak and possibly influenced by two outliers (vaquita and baiji). It also does not find a parallel in the evolution of menopause in humans, which have considerably less sexual dimorphism in body size compared to their hominid relatives orangutans and gorillas, and comparable sexual dimorphism compared to chimpanzees, all which of course do not have menopause as humans do. Both the hypothesis and result about sexual dimorphism in body size do not seem very robust, given the small sample size and lack of accounting for social factors.

We thank the reviewer for their comment. We also agree with the reviewer that an analysis including social data would be interesting but concur that the data are not available at the moment. In response to these comments, we have rewritten this section to (1) acknowledge the assumption underlying this analysis and (2) include some new published analysis supporting this hypothesis in killer whales (lines

152-171). We have also run an additional analysis to demonstrate that removing vaquita and baiji from the analysis makes no qualitative difference to the result (table S5). We also reduced the emphasis on the comparisons to humans, and highlight that humans and toothed whales do not share the same sexual competitive regimes in the discussion (lines 340-346).

The second set of supporting results tests whether the greater opportunities for intergenerational help in species with menopause are associated with either increased female reproductive rate relative to an ancestor without menopause, or more costly offspring relative to species without menopause. Using demographic simulations, the authors find that reproductive rates are slower in the species with menopause compared to the ancestral condition. They also find that offspring in species with menopause were not more costly based on proxy measures (age at maturity and adult body size) that did not differ among toothed whale species depending on the presence of menopause.

The third set of supporting results finds that by extending the lifespan without also extending the reproductive lifespan, species with menopause avoid potential costs in the form of intergenerational competition for resources. Specifically, (1) mother-daughter reproductive overlap did not differ according to the presence of menopause; and (2) in species with menopause, simulations indicate that mother-daughter reproductive overlap would be greater if females were to continue reproducing. While the patterns described are persuasive and robust, these findings also seem foreseeable given the finding that reproductive lifespans relative to body size remain typical in the species with menopause.

Originality and significance

The authors do a fine job of framing the study's significance by drawing parallels to the evolution of menopause in humans, but I have a few minor quibbles. The take-home statement of significance occurs on lines 318–319: “our results show that the same pathway that lead to the evolution of menopause in humans also lead to the evolution of menopause in toothed whales” Note that both instances of “lead” in this sentence should instead be “led,” the past tense of “lead” (the similar sentence in the abstract on line 27 correctly uses “led”). I found this framing about “the same pathway” a bit odd—as though there is a single, unambiguous, widely accepted set of factors that explains the evolution of menopause in humans. I suppose the authors are using “pathway” to refer to finding strong support for “live long” as opposed to “stop early” in both humans and toothed whales. But elsewhere in the manuscript (line 40), the authors state that “the pathways and mechanisms by which menopause evolves remain debated,” and the most recent review article cited in this section concludes with “The question of why prolonged life after the cessation of fertility has evolved in some species has not been fully answered” (Croft et al. 2014, p. 414). Thus, the assertion that the evolution of menopause in humans and toothed whales occurred “by the same pathway” may be a bit too sweeping.

We had used the term “pathway” to refer to the finding of strong support for the “live long” as opposed to the “stop early” hypothesis. We have now reworded this sentence in the discussion (lines 640-346). We thank the reviewer for pointing out this confusion.

Data & methodology: validity of approach, quality of data, quality of presentation

I had no major concerns about the data, methodology, or the validity of the approach. As someone who does not work with toothed whales, I am not familiar with the age determination method based on teeth or the reproductive lifespan determination method based on ovarian corpora. But they appear to be well validated.

The quality of the presentation was sufficient (Figure 1 was particularly nice). In Figure 2, I think it would help to use a bit more visual contrast between the line ranges representing the 50% and 95% credible intervals (e.g., by making the 50% intervals slightly thicker or darker).

We have made the 50% interval thicker and the 95% interval lighter as per the reviewers useful suggestion (figure 2). We note that through this exercise, we noticed that the 95% interval for one species was cut off in both facets a and b, which has now been corrected (and therefore the y axis limit has now changed slightly).

I had many issues with the supplementary figures.

- Fig. S1: The labels with species common names are cut off. The filled horizontal bars are redundant, as you could just show the points and line ranges with the same color coding. The salmon/teal color coding for used here to show presence or absence of menopause differs from the red/blue coding of figures in the manuscript. The color coding is not explained in a legend or in the caption. The figure is supposed to show that species with menopause have smaller female/male body size ratios, but the visual presentation is not convincing (the menopause species seem to fall squarely within the range of other species) and suggest that the result may be driven primarily extreme sexual dimorphism in the opposite direction in the vaquita and baiji.
- Fig. S2: Uninformative axis labels (e.g., “log.size” vs. “Size (log)”) with no units and inconsistent capitalization. What do the fitted lines and gray shaded regions in each panel show? Why do these seem to fit the data quite poorly by passing well below most of the points (e.g., in panel a there are 6 reddish points below the line and at least 21 above the line)? The caption says “Females of species with menopause do not produce larger female or male offspring given their size,” but the figure does not seem to show this. Instead, it seems to show offspring age at maturity given the female’s body size. Same issue as noted above with salmon/teal color coding.
- Fig. S3: Many of the same issues as in Fig. S2.
- Fig. S4: I couldn’t find a reference to this figure anywhere in the text. Why is there a gray entry for NA in the legend? What does “(age Z)” mean in the x-axis title? The y-axis and caption are not entirely consistent (is it age at last reproduction [e.g., a mean], or oldest known reproduction?)

There are two documents called Supplementary 2. The supplementary document about phylogeny, size, etc. has a very informal style with many typos—it looks and reads like a rough draft and should be proofread more carefully. There are citations in this document but no bibliography. I will admit that I did not get far into this document with careful reading.

We apologise for the errors in the supplementary text and figures in our first submission. All supplementary figures have been remade, including incorporating the changes suggested by the reviewers. Please note that due to the incorporation of additional supplementary figures numbering has now changed (previous S1 = current S2; previous S2 = current S3; previous S3 = current S4), and previous figure S4 has now been removed because it is not referred to in the text. We have re-written the Additional Data Explanation (supplementary 3, now correctly labelled) to formalise the style and remove the errors.

With regard to figure S3 panel b (previously S2). We have investigated the visually 'poor fit' of the model. Two factors contribute to this- the first is 'uncertainty' around estimates of age at maturity (in this case likely to represent true variation around the age at maturity). This uncertainty is carried through the model, so even where points are 'above' the fitted area in most cases they are still within the lower edge of the age at maturity range for the species. Secondly, the models include phylogenetic autocorrelation. Many of the species in the cloud of points with (log) Sizes between 5 and 5.5 are closely related, and therefore have less influence over the fit of the model. Removing the uncertainty and/or the phylogenetic affect improves the visual fit of the model, but we argue is not justified either statistically or biologically. To improve clarity, we now include species names as labels on the figures S3 and S4, and include a clearer description of the role of error in the figure legends.

Overall, this was an exciting, high-quality study with a couple of compelling topline results followed by an assortment of less interesting supporting results that seem to confirm what seem like the logical consequences of living longer without changing the reproductive lifespan, or test predictions about intergenerational help and conflict using indirect proxies.

Referee #4 (Remarks to the Author):

Review of: The Evolution of Menopause in Toothed Whales

This paper brings together: (1) an impressive set of data, collated across a variety of sources; (2) cutting edge demographic analysis methods and; (2) sophisticated use of simulation, to make inferences about the most likely causes of the evolution of menopause in toothed whales.

The paper is well executed and the review of different hypotheses is very clear. The combined evidence and simulations come down in favour of the 'live-long' hypothesis: it is the post-reproductive lifespan that was most likely selected for, rather than the 'stopping-early' hypothesis in which reproductive cessation is selected for. The causal argument put forward is that post-reproductive lifespans simultaneously allow older females to help provision and / or support younger generations while avoiding reproductive competition with younger females.

I appreciated the effort to write this paper for a broad audience, though I found myself wanting more methodological details peppered throughout the text rather than having to wait for the methods section to learn about the results in detail. Key definitions, for example the definition of grandmother overlap, could have been provided when first introduced as they are not necessarily as intuitive as the text implies. This seems important if the comparison to humans is to remain a prominent feature of the paper (see comments below).

We have added more detail of how key metrics were calculated in the main text (lines 130-135, 229-233)

In places, the effect sizes seem relatively small for such long-lived species, e.g. less than one year difference in relative grand offspring years between species with and without menopause (Lines 140-142), so I found myself wanting more help interpreting the importance of the different results relative to each other.

We have clarified this text to emphasise that, for a given species, a unit of 1 in this analysis represents 'one age at maturity' (so if age at maturity = 10, 1 relative grandoffspring years = 10 years). (lines 143-144)

My major comment on the paper is that the inference from these analyses to humans is probably stretched beyond what is justified by either the present analyses or the evidence for human menopause evolution. The paper is framed around the usefulness of the presented data for understanding the human case. Really the whales are interesting enough in their own right (and I say this a person who researches humans!), so I would suggest toning down or editing the arguments about humans. I have two main reasons for this critique.

First, as of course the authors know, we don't have phylogenetically close species comparisons for

humans. However, while certain demographic and life-history features that differ across toothed whale species with and without menopause are potentially illuminating and / or consistent with those observed in humans, there are many other unobserved factors that make a straightforward comparative inference about major evolutionary forces of human menopause / post-reproductive lifespan evolution difficult.

Second, important causal mechanisms - like reproductive overlap across generations and indeed, the assumption of natural survival and fertility in humans - are almost certainly culturally regulated. All human populations have (highly variable) rules about, for example, who can marry / have sex with whom, in what place and at what age – and so evidence that there is more or less reproductive overlap in humans will vary by cultural and ecological context (see Mace & Alvergne 2012; Koster et al 2019) – there isn't really a species-wide measure that could serve as a comparator to the toothed whale data. Indeed, menopause progression itself appears to be variable (within constraints) across human populations, progressing differently in populations with different kinship structures (see for example, Snopkowski, Moya & Sear 2014). There is also a substantial debate in the human literature, so claims that the evidence for one or other evolutionary hypothesis for human menopause evolution is “overwhelming” is itself as yet too strong.

For these reasons I don't find it persuasive to argue that the mechanisms driving the evolution of menopause are even likely the same in toothed whales and humans. Unfortunately we simply don't have the evidentiary basis to make such a comparative claim. That's not to say that human researchers shouldn't consult this paper when thinking about the evolution of menopause / post-reproductive lifespan. They absolutely should.

We thank the reviewer for their positive opinion of our work and their useful comments. In response to this comment and those from other reviewers we have refocused the manuscript to focus less on the comparison between humans and toothed whales. Key changes include: in the introduction more nuance has been added to the discussion of evolution of menopause in humans (lines 41-47); in the discussion the comparisons to humans has been toned-down and partly removed (lines 274-282); discussion last paragraph has been completely rewritten (lines 340-346).

Mace, R., & Alvergne, A. (2012). Female reproductive competition within families in rural Gambia. *Proceedings of the Royal Society B: Biological Sciences*, 279(1736), 2219-2227.

Koster J. et al. (2019) Kinship ties across the lifespan in human communities." *Philosophical Transactions of the Royal Society B* 374.1780: 20180069.

Snopkowski, K., Moya, C., & Sear, R. (2014). A test of the intergenerational conflict model in Indonesia shows no evidence of earlier menopause in female-dispersing groups. *Proceedings of the Royal Society B: Biological Sciences*, 281(1788), 20140580.

Reviewer Reports on the First Revision:

Referees' comments:

Referee #1 (Remarks to the Author):

Thanks to the authors for thoroughly responding to my concerns. I have no further comments on the manuscript, except that it will make a very fine contribution to the literature

Referee #2 (Remarks to the Author):

The authors have done a great job of responding to my initial review. I appreciate the efforts they have gone to in addressing all reviewer's comments.

I had a couple of very minor points.

I think referring back to the species with a dot next to their names in Figure 1a would be a helpful addition to the legend of Figure 2, as a reminder these are the subset of species with precise reproductive lifespan data.

Line 321-323. I had to read this sentence a couple of times before I got the meaning. I would suggest adding commas after 'ages' and 'harm':

When female relatedness to her local group increases as she ages, older females are under strong selection to help and reduced selection to harm, which can favour the evolution of menopause²⁶.

Overall this study is an exciting and comprehensive into the evolution of the menopause. My congratulations to the authors.

Referee #3 (Remarks to the Author):

The authors have done an excellent job of addressing the comments that I and the other reviewers raised in this revised version of the manuscript. The reduced emphasis on direct parallels to the evolution of menopause in humans has removed my remaining concerns about how the results are framed in terms of their significance. Table 1 is also a nice addition, and the Supplementary Materials are now much more clearly organized and written. I feel this study will be of significant interest to a wide audience, and that it is suitable for publication in Nature.

Referee #4 (Remarks to the Author):

Many thanks to the authors for their revision of the paper, in particular for toning down the strength of claims for human menopause evolution. I found the revision even clearer and more compelling as a result. I have no further comments to report.

Referee #5 (Remarks to the Author):

The authors presented a comparative study on the evolution of menopause in toothed whales, this taxonomic group is especially suited for a comparative analysis because menopause evolved independently multiple times in this group and allowed the authors to test several classical predictions related to the evolution of menopause. I enjoyed reading this manuscript as it was well structured and associated with robust analyses. I believe this article is of broad interest and will considerably increase our understanding of the evolution of menopause which makes it worth publishing in Nature.

I think that the results of this article are quite robust and I did not spot any major issue related to the analyses. I will hereafter detail my specific comments regarding the different predictions tested in this manuscript. I have mainly some comments regarding the demographic models used by the authors which I believe are quite minor and will not change the outcome of those analyses.

First the authors demonstrated that menopause species were associated to extended lifespans but no changes in reproductive lifespans compared to non-menopause species. Regarding the survival models fitted for adults, it would be nice to add the figures representing the fit of the Gompertz model for each population to see whether the use of parametric models such as Gompertz will not under or over-estimate the longevity. The authors already added that for the fertility models and they should add those figures for the survival models. Moreover, a supplementary table giving the outcome parameters and the posterior intervals associated for both fertility and survival models should be added. From what I understand the main raw data used for survival analysis consisted of mass-stranding or hunted individuals which could be considered as snapshot of the age distribution of alive individuals. I was wondering if the authors encountered data that are closer to the age at death distribution such as random strandings of dead individuals and if yes how did the authors performed the analyses with those data? It is not mentioned right now in the methods.

With the second prediction the authors tested whether menopause species spent more time alive with their granddaughters, to perform those kinship analysis the authors needed the whole age-specific demographic trajectories for each population. I understand that it is particularly hard to have accurate juvenile survival for mammals and even more for whales, but I found the choice of modelling juvenile independently juvenile survival and adult survival questionable. For 23 species, the authors had sufficient data to model juvenile survival and did so using a reverse Gompertz model. My issue here is that by doing so it is possible that the authors can end up with a correction for two different growth

rates, one for the adult sample and one for the juvenile sample which is not correct because population growth is a population parameter. I did not understand why in this case the authors did not consider the whole age distribution using a single bathtub model such as Siler that will not end up correcting for different population growth rates for that sample of species. Relating to that issue, the authors say lines 616 “exploring the parameter space to find the value of f resulting in a stable population ($\lambda=1$)”. I think the authors must be consistent on how they consider and correct for population growth rate. Either they correct for the same population growth rate for juvenile survival, adult survival and fertility or they do not, and consider that all the populations are stable.

Figure 2a is interesting and I think that only describing it using the mean of each group is reducing the picture. I agree that on average grandmother years are higher for menopause species however some non-menopause species are also characterised by high values of grandmother years such as harbour porpoises and sperm whales. This is interesting because it means that some species with higher presence of grandmothers, and then the possibility of more intergenerational help, do not show menopause and it also goes well with the discussion part on that (Lines 314-339).

Lines 134 of supplementary 3 “Survival from birth to maturity (s_0) shows a negative relationship with age at maturity in this sample 134 (post. mean = -0.16, 95%CI = -0.32-0.01; figure SE4)” and also mentioned lines 603 and 604 of the main text but the figure S4E represents the juvenile survival in function of size not the age at maturity.

The third prediction tested if menopause species are associated with higher intergenerational help. They used age at maturity and adult size as a measure of the reproductive investment and found no evidence of such effect. I do not know much about growth in whales but in mammals the form of growth curves can be quite variable. The authors here are only using the end point of growth curves looking at age and size at maturity which offer only a reduced view of the investment needed for example some earlier development period could be more critical with more investment needed. Might be worth discussing it to balance this absence of result, although I would assume that the amount of growth data needed to test this are lacking right now.

Referee #6 (Remarks to the Author):

This study uses data on the demography of 32 toothed whale species to test differing theories for the evolution of post-reproductive survival, which has evolved independently four times in five species. It is an exciting dataset and exciting results on a question of major interest. I do however have a number of important concerns with the manuscript in its current form, moreso than the other reviewers. Also, sorry I'm late to the game – I feel bad coming in with important comments at this stage.

Hypotheses considered

My most important concern regards the framing of the relevant hypotheses. The article as structured lays out 5 hypotheses, the latter three of which can be seen as more specific versions of the first two.

However, these are by no means the only potential hypotheses. Here, for example, are two which the data test, and which are consistent with the findings in the end: (1) Longer lifespan evolves in cases where pod success depends more strongly on wisdom and experience of the oldest individuals, regardless of the direct effects on specific offspring and grandoffspring. (Group selection is not usually a strong force, but the conditions are close to perfect for at least some in both tooth whale pods and in early humans.) This clearly predicts the evolution of long lifespan rather than short reproduction, as found. (2) There are different physiological costs/constraints/trade-offs around reproductive and somatic aging, and these result in a mismatch between the timing of menopause and mortality. Under the right social conditions, lifespan might be extended at low cost, but reproductive span is more costly. This also predicts the finding of extended lifespan, not shorter reproduction.

I'd like to expand on the second of these hypotheses a bit. It has been proposed and argued for by a number of authors, notably Nichols et al. *Biology Letters* 2016 and Cohen *Biological Reviews* 2004. It is also supported by recent theoretical work by Moorad (e.g. Moorad and Ravindran *American Naturalist* 2022) showing the importance of different rates of selection on different aging components. To my mind, the interesting explanation for menopause rests at the intersection of the physiological constraints and the social/kin selection-based theories: the presence of a mismatch in how selection works on somatic and reproductive aging creates an opportunity for the right social conditions to select for extended lifespan, but not extended reproductive span, in females, at minimal cost. This is exactly what the earlier papers predicted, and what is found here. I thus think the selection of hypotheses chosen by the authors may well lead to conclusions that are erroneous or only partially correct.

Males

An alternative approach to considering whether menopause evolves based on extension of lifespan or reduction of reproduction is to compare male and female lifespan. Presumably, these data were also available. Why were they not queried too? The prediction would be that female: male lifespan ratio would be substantially higher in species with menopause. This is certainly true in short-finned pilot whales, though not in humans. See, for example,

- Moorad, J. A., & Walling, C. A. (2017). Measuring selection for genes that promote long life in a historical human population. *Nature ecology & evolution*, 1(11), 1773-1781.
- Tuljapurkar, S. D., Puleston, C. O., & Gurven, M. D. (2007). Why men matter: mating patterns drive evolution of human lifespan. *PloS one*, 2(8), e785.

Statistics

Overall, the authors use statistics impressively and well. However, there are a couple specific errors that I think do have a major impact on the interpretation of results. In Figure 2, the distributions of the two key variables (Grandmother years and Reproductive overlap) look like they may well have bimodal distributions. There are a set of species that are very low on both, and another set of species that are much higher on both. All the menopausal species are high on both. If these are continuous variables with a normal distribution, the analyses conducted are likely appropriate, but if they reflect two "modes" of life, all we are saying is that menopausal species are more likely to be in Mode 2 (higher values on both). The sample size is limited to draw a clear conclusion on the correct analysis, but this

concern means there is substantial uncertainty about what is occurring here. There may be some social preconditions for evolution of menopause, but only some species with these conditions evolve menopause, for example. This would also be interesting, but is quite different from the conclusion drawn regarding Fig. 2A: that menopausal species are systematically different than non-menopausal species.

My larger concern with the statistics is related, and regards Fig. 2B. Here, the authors conclude that there is no difference between menopausal and non-menopausal species. They are certainly right that there is no evidence of a difference, but the problem is that there isn't really much evidence one way or the other. The error bars in Fig. 2B are much larger than in 2A, meaning there is substantial uncertainty in the estimation of reproductive overlap for most species. However, the point estimates are just as distinct as in 2A, and follow the same pattern. This suggests to me that the analysis in 2B is substantially underpowered, but would likely in the end emerge significant if there were more power (based on it being largely the same species estimated to be high versus low in 2A and 2B). What is certainly clear is that the authors can't conclude that there is no difference in reproductive overlap – they can't conclude one way or the other. And this of course has a large impact on the hypotheses...

Related to uncertainty questions, I think there is substantial uncertainty in the classification of species as menopausal or not. There's no reason to think this trait should be dichotomous; furthermore, the ability to detect it depends a lot on demography. Under high-mortality conditions, a species might appear non-menopausal, and indeed, for many species, the current conditions in the Anthropocene are likely leading to exactly the kinds of conditions that might hide that a species is menopausal, particularly when estimated from small sample sizes and/or from a single population. The authors did an excellent job of building uncertainty into the Bayesian models and letting it propagate, but I don't believe they built in uncertainty about menopausal status itself. As with my previous comment, we should be cautious about interpreting absence of evidence for menopause as evidence of the absence of menopause.

A final, much more minor statistical point: the 5 menopausal species fall within a narrow size band, with most of the non-menopausal species much smaller. Although I don't have any good reason to suspect it, this at least raises the remote possibility that there is something specific about the biology of intermediate-sized whales that leads to these differences, and that the linear models in Fig. 1 are not appropriate. (It's unfortunate that the 5 menopausal species weren't scattered along the size distribution.) But without a clearer reason to worry about this, I'm not overly concerned.

Tautological hypotheses?

Hypothesis 3 is tested via greater grandparent years in menopausal species. But isn't this inevitable given what is observed in Fig 1B-C? That is, given that we have supported hypothesis 1 rather than hypothesis 2, won't we necessarily also support hypothesis 3 by finding greater grandparent years in menopausal species, which live longer than expected? If this is not tautological, it should clearly be shown why, since I think one other reviewer also noted it.

Reproductive overlap does not necessarily relate to intergenerational harm

Hypothesis 5 (Intergenerational harm) is tested by looking for reproductive overlap. The absence of greater reproductive overlap in menopausal species is taken as evidence for intergenerational harm which prevents the evolution of such overlap. (As noted above, I suspect there are actually differences in reproductive overlap, though the data aren't clear.) But regardless of the results, it's not correct to assume that the two are related. For example, if there is greater competition between pods than within pods, it might be beneficial to grow the size of one's pod rather than worry about competition. This likely depends on ecological conditions, population trends, pod size, and many factors.

A few more minor points:

- Ovarian activity as a surrogate for reproduction may confound physiology and behavior in this context
- Fig. 1b: was the line derived from all species or only non-PrR ones? Make this clear in the legend please. (I think this was intended to be included but the relevant sentence has a typo and it's not clear.)
- In Fig. 2b, PrR species are higher than the line in all cases. Individually this may not be significant, but is it collectively? Do we have the power to rule out an effect, though almost certainly more modest, for reproductive span?
- There is an assumption that social system (e.g. bisexual philopatry) could drive the evolution of menopause, but couldn't the arrow go the other way? Do we know how stable social systems like this are over evolutionary time? In humans, not that stable.
- What about Asian elephants? Aren't they worth at least a mention here?

Author Rebuttals to First Revision:

Referees' comments:

Referee #1 (Remarks to the Author):

Thanks to the authors for thoroughly responding to my concerns. I have no further comments on the manuscript, except that it will make a very fine contribution to the literature

We thank the reviewer for their useful comments during the review process and positive evaluation of the manuscript.

Referee #2 (Remarks to the Author):

The authors have done a great job of responding to my initial review. I appreciate the efforts they have gone to in addressing all reviewer's comments.

I had a couple of very minor points.

I think referring back to the species with a dot next to their names in Figure 1a would be a helpful addition to the legend of Figure 2, as a reminder these are the subset of species with precise reproductive lifespan data.

Addition made as suggested line 281.

Line 321-323. I had to read this sentence a couple of times before I got the meaning. I would suggest adding commas after 'ages' and 'harm':

When female relatedness to her local group increases as she ages, older females are under strong selection to help and reduced selection to harm, which can favour the evolution of menopause²⁶.

Change made as suggested lines 356-357

Overall this study is an exciting and comprehensive into the evolution of the menopause. My congratulations to the authors.

We thank the reviewer for their thoughtful comments and positive evaluation of the manuscript.

Referee #3 (Remarks to the Author):

The authors have done an excellent job of addressing the comments that I and the other reviewers raised in this revised version of the manuscript. The reduced emphasis on direct parallels to the evolution of menopause in humans has removed my remaining concerns about how the results are framed in terms of their significance. Table 1 is also a nice addition, and the Supplementary Materials are now much more clearly organized and written. I feel this study will be of significant interest to a wide audience, and that it is suitable for publication in Nature.

We thank the reviewer for their insightful comments and suggestions as part of the review process and their supportive evaluation of the manuscript.

Referee #4 (Remarks to the Author):

Many thanks to the authors for their revision of the paper, in particular for toning down the strength of claims for human menopause evolution. I found the revision even clearer and more compelling as a result. I have no further comments to report.

We thank the reviewer for their previous comments and their positive evaluation of the manuscript.

Referee #5 (Remarks to the Author):

The authors presented a comparative study on the evolution of menopause in toothed whales, this taxonomic group is especially suited for a comparative analysis because menopause evolved independently multiple times in this group and allowed the authors to test several classical predictions related to the evolution of menopause. I enjoyed reading this manuscript as it was well structured and associated with robust analyses. I believe this article is of broad interest and will considerably increase our understanding of the evolution of menopause which makes it worth publishing in Nature.

I think that the results of this article are quite robust and I did not spot any major issue related to the analyses. I will hereafter detail my specific comments regarding the different predictions tested in this manuscript. I have mainly some comments regarding the demographic models used by the authors which I believe are quite minor and will not change the outcome of those analyses.

First the authors demonstrated that menopause species were associated to extended lifespans but no changes in reproductive lifespans compared to non-menopause species. Regarding the survival models fitted for adults, it would be nice to add the figures representing the fit of the Gompertz model for each population to see whether the use of parametric models such as Gompertz will not under or over-estimate the longevity. The authors already added that for the fertility models and they should add

those figures for the survival models. Moreover, a supplementary table giving the outcome parameters and the posterior intervals associated for both fertility and survival models should be added.

We thank the reviewer for this suggestion which we now include the figures, and a table of parameter estimates as suggested (Supplementary 2).

From what I understand the main raw data used for survival analysis consisted of mass-stranding or hunted individuals which could be considered as snapshot of the age distribution of alive individuals. I was wondering if the authors encountered data that are closer to the age at death distribution such as random strandings of dead individuals and if yes how did the authors performed the analyses with those data? It is not mentioned right now in the methods.

This is an interesting point that we did consider during our analysis. 27 of our 269 datasets come from sources that are from “ad-hoc” stranding events- i.e. collated data from individual stranding events. The 27 datasets are from 8 species and in only one species (Dwarf Sperm Whales) do stranding data consist of the only data source for a species.

Contrary to our expectation when we started this project there is no indication that these datasets are biased towards particular ages of death, indeed in all cases they closely resemble the age-distributions of datasets from other sources (supplementary 2). Excluding these datasets makes little difference to lifespans estimates for these species, but to confirm this we re-ran our key analyses without including these datasets in the estimates of lifespan. There was no quantitative and very little qualitative difference to our results when these datasets are excluded (table S5). We now highlight these datasets in our manuscript (supplementary 2), and report a version of our results excluding these datasets (table S5).

With the second prediction the authors tested whether menopause species spent more time alive with their granddaughters, to perform those kinship analysis the authors needed the whole age-specific demographic trajectories for each population. I understand that it is particularly hard to have accurate juvenile survival for mammals and even more for whales, but I found the choice of modelling juvenile independently juvenile survival and adult survival questionable. For 23 species, the authors had sufficient data to model juvenile survival and did so using a reverse Gompertz model. My issue here is that by doing so it is possible that the authors can end up with a correction for two different growth rates, one for the adult sample and one for the juvenile sample which is not correct because population growth is a population parameter. I did not understand why in this case the authors did not consider the whole age distribution using a single bathtub model such as Siler that will not end up correcting for different population growth rates for that sample of species.

We thank the reviewer for this useful suggestion. As suggested by the reviewer we added a bathtub term to our Gompertz survival model (we used the Gompertz rather than Siler model for consistency with the rest of the analysis)- and used this extended model to run our juvenile analysis over the whole age distributions. We incorporated these improved estimates of juvenile survival into the demographic analyses (e.g. figure 2)- there is no qualitative difference to the results (and little quantitative difference).

The details of the new analysis are in lines 662-663 Supplementary 3 lines 111-137 and the relevant results have been updated throughout the manuscript, supplementaries and figures.

Relating to that issue, the authors say lines 616 “exploring the parameter space to find the value of f resulting in a stable population ($\lambda=1$)”. I think the authors must be consistent on how they consider and correct for population growth rate. Either they correct for the same population growth rate for juvenile survival, adult survival and fertility or they do not, and consider that all the populations are stable.

We apologise for the lack of clarity here. With this particular analysis we are -as the reviewer states – considering that each population is stable. We explore the parameter space to find the appropriate value of f given that we ‘know’ the population is stable. We derive estimated parameters of survival and fertility under a stable population (by setting the r parameter to 1) and then use these values from a stable population to calculate the required baseline rate of reproduction to maintain that stable population size. We have reworded this section to make this clear and we thank the reviewer for highlighting this point (lines 673-677).

Figure 2a is interesting and I think that only describing it using the mean of each group is reducing the picture. I agree that on average grandmother years are higher for menopause species however some non-menopause species are also characterised by high values of grandmother years such as harbour porpoises and sperm whales. This is interesting because it means that some species with higher presence of grandmothers, and then the possibility of more intergenerational help, do not show menopause and it also goes well with the discussion part on that (Lines 314-339).

We thank the reviewer for raising this point and we have added this to the Discussion lines 348-350

Lines 134 of supplementary 3 “Survival from birth to maturity (s_0) shows a negative relationship with age at maturity in this sample 134 (post. mean = -0.16, 95%CI = -0.32-0.01; figure SE4)” and also mentioned lines 603 and 604 of the main text but the figure S4E represents the juvenile survival in function of size not the age at maturity.

We apologise for the confusion figure SE4 has been replaced with the correct figure (with the new values from the reanalysis of survival to maturity).

The third prediction tested if menopause species are associated with higher intergenerational help. They used age at maturity and adult size as a measure of the reproductive investment and found no evidence of such effect. I do not know much about growth in whales but in mammals the form of growth curves can be quite variable. The authors here are only using the end point of growth curves looking at age and size at maturity which offer only a reduced view of the investment needed for example some earlier development period could be more critical with more investment needed. Might be worth discussing it to balance this absence of result, although I would assume that the amount of growth data needed to

test this are lacking right now.

We thank the reviewer for raising this interesting point and we have added a discussion of this to the manuscript (lines 216-217). We agree that it would be interesting to explore this further in our analysis but as suggested by the reviewer, unfortunately, the data do not exist to test this.

Referee #6 (Remarks to the Author):

This study uses data on the demography of 32 toothed whale species to test differing theories for the evolution of post-reproductive survival, which has evolved independently four times in five species. It is an exciting dataset and exciting results on a question of major interest. I do however have a number of important concerns with the manuscript in its current form, moreso than the other reviewers. Also, sorry I'm late to the game – I feel bad coming in with important comments at this stage.

Hypotheses considered

My most important concern regards the framing of the relevant hypotheses. The article as structured lays out 5 hypotheses, the latter three of which can be seen as more specific versions of the first two. However, these are by no means the only potential hypotheses. Here, for example, are two which the data test, and which are consistent with the findings in the end: (1) Longer lifespan evolves in cases where pod success depends more strongly on wisdom and experience of the oldest individuals, regardless of the direct effects on specific offspring and grandoffspring. (Group selection is not usually a strong force, but the conditions are close to perfect for at least some in both tooth whale pods and in early humans.) This clearly predicts the evolution of long lifespan rather than short reproduction, as found. (2) There are different physiological costs/constraints/trade-offs around reproductive and somatic aging, and these result in a mismatch between the timing of menopause and mortality. Under the right social conditions, lifespan might be extended at low cost, but reproductive span is more costly. This also predicts the finding of extended lifespan, not shorter reproduction.

I'd like to expand on the second of these hypotheses a bit. It has been proposed and argued for by a number of authors, notably Nichols et al. *Biology Letters* 2016 and Cohen *Biological Reviews* 2004. It is also supported by recent theoretical work by Moorad (e.g. Moorad and Ravindran *American Naturalist* 2022) showing the importance of different rates of selection on different aging components. To my mind, the interesting explanation for menopause rests at the intersection of the physiological constraints and the social/kin selection-based theories: the presence of a mismatch in how selection works on somatic and reproductive aging creates an opportunity for the right social conditions to select for extended lifespan, but not extended reproductive span, in females, at minimal cost. This is exactly what the earlier papers predicted, and what is found here. I thus think the selection of hypotheses chosen by the authors may well lead to conclusions that are erroneous or only partially correct.

Males

An alternative approach to considering whether menopause evolves based on extension of lifespan or reduction of reproduction is to compare male and female lifespan. Presumably, these data were also available. Why were they not queried too? The prediction would be that female: male lifespan ratio would be substantially higher in species with menopause. This is certainly true in short-finned pilot whales, though not in humans. See, for example,

- Moorad, J. A., & Walling, C. A. (2017). Measuring selection for genes that promote long life in a historical human population. *Nature ecology & evolution*, 1(11), 1773-1781.
- Tuljapurkar, S. D., Puleston, C. O., & Gurven, M. D. (2007). Why men matter: mating patterns drive evolution of human lifespan. *PloS one*, 2(8), e785.

We thank the new reviewer for raising these three points, which we respond to individually below.

1. **Focus on inclusive fitness.** *The reviewer is correct to point out that in this manuscript, we focus on inclusive fitness/kin selected explanations of the evolution of menopause. Our motivation here is to test the key hypotheses proposed to explain the evolution of menopause (manuscript reference [5]). These hypotheses have focused on kin-selected explanations (mother hypothesis ([13,27]), grandmother hypothesis ([13-15]) and the reproductive conflict hypothesis ([28,36]). These hypotheses have received empirical support in humans and killer whales ([6,7,22-25, 37]). What is missing from the literature is an analysis of how menopause evolved (by extending the total lifespan or shorting the reproductive lifespan), which is the key result in this manuscript. In response to the reviewer we have added new text to justify our focus on inclusive fitness throughout the manuscript (lines 41-44, 124-137). We also agree with the reviewer that it is important to consider and acknowledge alternative explanations and in response to the reviewer's comments, we have included new text that discuss other theories in a new paragraph in the Discussion (lines 375-382) and elsewhere (lines 98-101).*
2. **The role of physiological constraints/trade-offs.** *We agree with the reviewer that physiological constraints and trade-offs (and selection on those trade-offs) are likely to be an important part of the story of the evolution of menopause. But, as the reviewer points out, a key challenge is to understand the social conditions that drive these changes- which is the focus of this manuscript. We now discuss the potential role of physiological trade-offs in the manuscript (lines 98-101).*
3. **Male driven menopause.** *We thank the reviewer for this suggestion. We have now included a new analysis to test the male-driven menopause hypothesis in toothed whales (lines 259-276) We first compare the Male:Female lifespan ratio in species with and without menopause and find that in species with menopause males have relatively shorter lives than in species without menopause. We then investigated male and female lifespans in species with menopause. We found that in species with menopause that, compared to males, females are more likely to survive to the age at female menopause and will live-longer once they reach that age. Taken together, we find no evidence for the Male-driven menopause hypothesis in toothed whales. We thank the reviewer for raising this, we think it adds an interesting and important extra dimension to our analysis.*

Statistics

Overall, the authors use statistics impressively and well. However, there are a couple specific errors that I think do have a major impact on the interpretation of results. In Figure 2, the distributions of the two key variables (Grandmother years and Reproductive overlap) look like they may well have bimodal distributions. There are a set of species that are very low on both, and another set of species that are much higher on both. All the menopausal species are high on both. If these are continuous variables with a normal distribution, the analyses conducted are likely appropriate, but if they reflect two “modes” of life, all we are saying is that menopausal species are more likely to be in Mode 2 (higher values on both). The sample size is limited to draw a clear conclusion on the correct analysis, but this concern means there is substantial uncertainty about what is occurring here. There may be some social preconditions for evolution of menopause, but only some species with these conditions evolve menopause, for example. This would also be interesting, but is quite different from the conclusion drawn regarding Fig. 2A: that menopausal species are systematically different than non-menopausal species.

We have performed two additional analyses to investigate this interesting point- both analyses show that neither grandmother years or reproductive overlap are binomodal, there do not appear to be two modes of life in the toothed whales, and that our model successfully captures the true distribution of observed data.

First, we plotted both the separate and combined distributions of the Grandmother Years and Reproductive Overlap distributions (figure S6). Neither are clearly bimodal, and in both cases the distributions of species with menopause is not clearly separated from that of species without menopause (lines 760,791). We find no evidence that there are two distinct modes of life in toothed whales.

Second, we performed posterior predictive checks on the models underlying the Grandmother Years analysis (figure 2a) and Reproductive Overlap analysis (figure 2b) to establish their ability to capture the observed distributions. In both cases, the posterior distributions capture the observed distributions (figure S7; lines, 758-759, 789-790)- indicating that our models reflect the observed data- and can therefore be used for reliable statistical inference.

My larger concern with the statistics is related, and regards Fig. 2B. Here, the authors conclude that there is no difference between menopausal and non-menopausal species. They are certainly right that there is no evidence of a difference, but the problem is that there isn't really much evidence one way or the other. The error bars in Fig. 2B are much larger than in 2A, meaning there is substantial uncertainty in the estimation of reproductive overlap for most species. However, the point estimates are just as distinct as in 2A, and follow the same pattern. This suggests to me that the analysis in 2B is substantially underpowered, but would likely in the end emerge significant if there were more power (based on it being largely the same species estimated to be high versus low in 2A and 2B). What is certainly clear is that the authors can't conclude that there is no difference in reproductive overlap – they can't conclude one way or the other. And this of course has a large impact on the hypotheses...

We have performed two additional analyses to confirm the ability of our modelling framework to detect a difference in reproductive overlap between species with and without menopause.

First, we investigate the power of our models by exploring the role of the uncertainty around our reproductive overlap estimates in driving our conclusion of an absence of an effect. We ran two tests to investigate if the absence of effect of the menopause parameter is because of the larger error around the reproductive overlap values (figure 2b) than the grandmother years values (figure 2a). a) in an analysis of just the point estimates without the error there is still no effect of menopause ($\beta = 0.34$, 95CI= -0.33 – 0.99; proportion of posterior menopause > no menopause = 0.81) and b) if the true reproductive overlap error is replaced with the (scaled) error from the grandmother years for that species there is still no effect of menopause in the model ($\beta = 0.35$, 95CI= -0.30 – 0.97; proportion of posterior menopause > no menopause = 0.81). These tests show that the difference in result between the Grandmother Years analysis (figure 2a) and the Reproductive Overlap analysis (figure 2b) is not simply explained by the greater error around the point estimates (lines 793-795).

Secondly, we explore the relative strength of models with and without a menopause parameter. If species with menopause had a greater reproductive overlap than species without menopause in an under-powered model, we would expect that the models with a menopause parameter would have a greater predictive power, even when the distribution of that parameter overlapped zero. To test this prediction, we used leave-one-out-cross-validation to compare the predictive power of models with a menopause parameter (figure 2b) and without a menopause parameter (table S7). Models with menopause as a parameter do not have a greater predictive power than models without a menopause parameter- indeed, there is weak evidence that they have a lower predictive power (table S7). The result of this supplementary analysis adds further support to our conclusions and indicates that the lack of a difference in reproductive overlap between species with and without menopause is not simply because our model is under-powered. This analysis is reported and described in the text lines 252-254, 792-793 and table S7.

Related to uncertainty questions, I think there is substantial uncertainty in the classification of species as menopausal or not. There's no reason to think this trait should be dichotomous; furthermore, the ability to detect it depends a lot on demography. Under high-mortality conditions, a species might appear non-menopausal, and indeed, for many species, the current conditions in the Anthropocene are likely leading to exactly the kinds of conditions that might hide that a species is menopausal, particularly when estimated from small sample sizes and/or from a single population. The authors did an excellent job of building uncertainty into the Bayesian models and letting it propagate, but I don't believe they built in uncertainty about menopausal status itself. As with my previous comment, we should be cautious about interpreting absence of evidence for menopause as evidence of the absence of menopause.

We thank the reviewer for raising this important point. Our data structure allows us to include the extent of post-reproductive life as a continuous variable rather than including menopause as a binary trait (1 or 0). We have performed an additional analysis using the Post Reproductive Representation values for the toothed whale species (where known) with the uncertainty around those values. Post Reproductive Representation measures the proportion of adult life years lived by post-reproductive females in a population- it is a continuous measure of the demographic importance of the post-reproductive female lifespan in a population. Replacing our binary measures of menopause with this continuous measure- and including the error around these measures – does not qualitatively change our results (table S5). This analysis incorporates both the uncertainty around menopause state and allows for a continuous rather than binary menopause state. We retain our binary classification in the main text but report this analysis of continuous measure both in the main text and supplementary material (lines 749-751, table S5).

A final, much more minor statistical point: the 5 menopausal species fall within a narrow size band, with most of the non-menopausal species much smaller. Although I don't have any good reason to suspect it, this at least raises the remote possibility that there is something specific about the biology of intermediate-sized whales that leads to these differences, and that the linear models in Fig. 1 are not appropriate. (It's unfortunate that the 5 menopausal species weren't scattered along the size distribution.) But without a clearer reason to worry about this, I'm not overly concerned.

We agree that the menopause species are not scattered evenly along the toothed whale sizes- though we note that some intermediate sized whales e.g. long-finned pilot whales, northern-bottlenose whales, do not have menopause. It would be interesting to investigate the coevolution of ecology and sociality with toothed whale size- but this is unfortunately beyond the scope of this analysis.

Tautological hypotheses?

Hypothesis 3 is tested via greater grandparent years in menopausal species. But isn't this inevitable given what is observed in Fig 1B-C? That is, given that we have supported hypothesis 1 rather than hypothesis 2, won't we necessarily also support hypothesis 3 by finding greater grandparent years in menopausal species, which live longer than expected? If this is not tautological, it should clearly be shown why, since I think one other reviewer also noted it.

This is an interesting point but here we use simulations to demonstrate that our hypothesis is not tautological.

Simulations demonstrate that it does not always follow that the life-long hypothesis leads to extended grandmother years. The extent of grandmother-grandoffspring lifespan overlaps are not dependent on adult-lifespan (figure 1a) and reproductive lifespan (figure 1b) but also on juvenile mortality, age at maturity and shape of adult survival curve (line 698-697 Changing any one of these other factors can lead to substantial changes in the amount of grandmother overlap.

For example, in the figure below, we compare the observed grandmother years of Short Finned Pilot Whales (0. Obs) simulated scenarios where (with lifespan and reproductive lifespan the same as the observed case): juvenile mortality rate is halved (1. Low Juve Mortality), 50% later age at maturity (2. Later Maturity) or a constant survival rate until halfway through adulthood without changing overall lifespan (3. Higher mid-life survival). In all three alternative scenarios, grandmother years go from being high (~1) to much more similar to the species without menopause (~0.5). The pattern is the same for other species.

These simulations show that hypothesis 3 does not necessarily follow from hypothesis 1- and is important to explicitly test hypothesis 3 to get a complete understanding of the evolution of menopause in toothed whales.

Response Figure 1. Simulated demography of Short Finned Pilot Whales.

Reproductive overlap does not necessarily relate to intergenerational harm

Hypothesis 5 (Intergenerational harm) is tested by looking for reproductive overlap. The absence of greater reproductive overlap in menopausal species is taken as evidence for intergenerational harm which prevents the evolution of such overlap. (As noted above, I suspect there are actually differences in reproductive overlap, though the data aren't clear.) But regardless of the results, it's not correct to assume that the two are related. For example, if there is greater competition between pods than within pods, it might be beneficial to grow the size of one's pod rather than worry about competition. This likely depends on ecological conditions, population trends, pod size, and many factors.

We agree that that reproductive overlap does not necessarily relate to intergenerational harm, and we have clarified the text to highlight that it simply provides the potential for intergenerational harm (lines 237-247) and that further research is necessary to quantify intergenerational harm in these systems- though as noted in the text in the only systems where this has been studied in detail -humans and killer whales – reproductive overlap has been found to be harmful (lines 227-228). As mentioned above, we also now discuss the potential role of intergroup conflict in the Discussion lines 375-382.

A few more minor points:

- Ovarian activity as a surrogate for reproduction may confound physiology and behavior in this context

We agree, unfortunately however, in most toothed whale species the data simply do not exist for a behavioural measure of age-specific reproduction. However, in support of our approach in species with both physiological and behavioural data we have previously found good agreement between the two: (Ellis et al. 2018)[main text reference 3] . We have edited the text to provide a discussion of this point 612-613.

- Fig. 1b: was the line derived from all species or only non-PrR ones? Make this clear in the legend please. (I think this was intended to be included but the relevant sentence has a typo and it's not clear.)

The reviewer is correct the line is for species without menopause. The legend has now been corrected (lines 115-116).

- In Fig. 2b, PrR species are higher than the line in all cases. Individually this may not be significant, but is it collectively? Do we have the power to rule out an effect, though almost certainly more modest, for reproductive span?

We thank the reviewer for this suggestion. We performed a similar test of model predictive power as for figure 2b (see response above). We found no evidence that the predictive power of models with a menopause parameter is higher than models without a menopause parameter, suggesting that our result is not due to a lack of power and represents a true absence of an effect. We have included this new analysis in the revised version of the manuscript (text lines 738-740; table S6).

- There is an assumption that social system (e.g. bisexual philopatry) could drive the evolution of menopause, but couldn't the arrow go the other way? Do we know how stable social systems like this are over evolutionary time? In humans, not that stable.

This is a very interesting point. Previous models of the evolution of menopause in humans and toothed whales have assumed that demographic patterns of dispersal and reproduction have driven the evolution of menopause rather than the other way around (e.g. main text references [5, 13-15, 28, 36]). However, as suggested by the reviewer, it is possible that there could be an interplay between the two. Unfortunately, we don't have the historical data to differentiate between these two causal pathways. However, we have now acknowledged this assumption in our manuscript (lines 365-366).

- What about Asian elephants? Aren't they worth at least a mention here?

We thank the reviewer for this suggestion. Asian elephants are now discussed as suggested lines 380-382

Reviewer Reports on the Second Revision:

Referees' comments:

Referee #5 (Remarks to the Author):

The authors have responded successfully to my various comments, so I have no further points to add. I thank the author for rerunning the different analysis taking my suggestions into account and I think that this article will have a major impact on menopause research.

Referee #6 (Remarks to the Author):

The authors have done a really impressive job of responding to my comments. I was reviewer #6 invited late in the process, and felt bad to leave major revisions at that stage; nonetheless, I did feel my comments had the potential to call into question the conclusions of the article. I was happy to see how seriously they were taken, and how much modeling was done to address them. While I am not yet 100% convinced, I am 90% convinced. I list the major points from my original review below, and how the new version does at addressing them:

- 1) Choice of hypotheses: Some reframing has made a huge difference in the interpretation and contextualization here. I am quite satisfied with this version.
- 2) Selection on males: The authors ran a number of analyses to test the hypothesis that selection on males generated correlated extension of lifespan in females. I was not completely convinced their predictions were precisely the right ones (it's tricky to puzzle out exactly what the expectation would be in this case), but I think the results are clear enough that we can eliminate this hypothesis as the key driver of the observed patterns in the data. I think it's good enough.
- 3) Bimodality of Grandmother Years and Reproductive Overlap: The authors did run some analyses to assess whether these variables might have bimodal distributions. They claim to show that the distributions are not bimodal, and that therefore their model is correct. However, in Figs S5 and S6, it is obvious to the eye that these distributions are both bimodal (or at least not unimodal). I can accept that there may be some uncertainty as to the true distribution, given limited sample sizes; perhaps we can't conclude for certain that they are bimodal. However, we surely cannot say with confidence that these are unimodal, and that then begs the question of whether we can compare menopausal and non-menopausal species while treating these variables as continuous. The second check (Fig. S5) appears to show that the demographic model recovers the true distributions. I must admit that I'm uncertain what that means in this case. Perhaps it is sufficient, but I think I would need to fully implement their model myself to feel confident I know what that means, and it's more time than I can put into this review to do that. My intuition remains the same: there is a group of (largely) small whales with very few grandmother years, and a group of (mostly) large whales with more grandmother years; all of the menopausal species are in the latter group, and the interpretation is tricky. I am not convinced any statistical modeling can solve this, I think it's simply a limitation of the number of species available for analysis. I think the authors have done all they can, except to acknowledge some uncertainty here.

4) Measurement/estimation error for reproductive overlap: Likewise, the authors have gone to great lengths to address my concern about the measurement error around reproductive overlap estimates for species. In this case I am somewhat more convinced, though not 100%. The authors do show that even with the point estimates there appears to be no distinction between menopausal and non-menopausal species, and that is as good as we'll get for now.

5) Classification of species as menopausal: The additional analysis is certainly sufficient here.

6) Tautological hypotheses: Very nice job showing I was wrong about the tautological hypotheses!

7) Reproductive overlap vs. intergenerational harm: the nuancing here is important, and sufficient.

Thank you!

So overall, only point #3 remains tricky. My feeling is that an acknowledgement of the uncertainty here and what that means for the conclusions should be sufficient. Even if the distribution is bimodal, it doesn't invalidate the whole article, it just makes the interpretation less clear. But the editor should weigh in.

Author Rebuttals to Second Revision:

Response to Referees

Referee #5 (Remarks to the Author):

The authors have responded successfully to my various comments, so I have no further points to add. I thank the author for rerunning the different analysis taking my suggestions into account and I think that this article will have a major impact on menopause research.

We thank the reviewer for their positive opinion of our manuscript.

Referee #6 (Remarks to the Author):

The authors have done a really impressive job of responding to my comments. I was reviewer #6 invited late in the process, and felt bad to leave major revisions at that stage; nonetheless, I did feel my comments had the potential to call into question the conclusions of the article. I was happy to see how seriously they were taken, and how much modeling was done to address them. While I am not yet 100% convinced, I am 90% convinced. I list the major points from my original review below, and how the new version does at addressing them:

We would like to thank the reviewer for their constructive comments and supportive evaluation of our manuscript.

- 1) Choice of hypotheses: Some reframing has made a huge difference in the interpretation and contextualization here. I am quite satisfied with this version.
- 2) Selection on males: The authors ran a number of analyses to test the hypothesis that selection on males generated correlated extension of lifespan in females. I was not completely convinced their predictions were precisely the right ones (it's tricky to puzzle out exactly what the expectation would be in this case), but I think the results are clear enough that we can eliminate this hypothesis as the key driver of the observed patterns in the data. I think it's good enough.
- 3) Bimodality of Grandmother Years and Reproductive Overlap: The authors did run some analyses to assess whether these variables might have bimodal distributions. They claim to show that the distributions are not bimodal, and that therefore their model is correct. However, in Figs S5 and S6, it is obvious to the eye that these distributions are both bimodal (or at least not unimodal). I can accept that there may be some uncertainty as to the true distribution, given limited sample sizes; perhaps we can't conclude for certain that they are bimodal. However, we surely cannot say with confidence that these are unimodal, and that then begs the question of whether we can compare menopausal and non-

menopausal species while treating these variables as continuous. The second check (Fig. S5) appears to show that the demographic model recovers the true distributions. I must admit that I'm uncertain what that means in this case. Perhaps it is sufficient, but I think I would need to fully implement their model myself to feel confident I know what that means, and it's more time than I can put into this review to do that. My intuition remains the same: there is a group of (largely) small whales with very few grandmother years, and a group of (mostly) large whales with more grandmother years; all of the menopausal species are in the latter group, and the interpretation is tricky. I am not convinced any statistical modeling can solve this, I think it's simply a limitation of the number of species available for analysis. I think the authors have done all they can, except to acknowledge some uncertainty here.

4) Measurement/estimation error for reproductive overlap: Likewise, the authors have gone to great lengths to address my concern about the measurement error around reproductive overlap estimates for species. In this case I am somewhat more convinced, though not 100%. The authors do show that even with the point estimates there appears to be no distinction between menopausal and non-menopausal species, and that is as good as we'll get for now.

5) Classification of species as menopausal: The additional analysis is certainly sufficient here.

6) Tautological hypotheses: Very nice job showing I was wrong about the tautological hypotheses!

7) Reproductive overlap vs. intergenerational harm: the nuancing here is important, and sufficient.
Thank you!

So overall, only point #3 remains tricky. My feeling is that an acknowledgement of the uncertainty here and what that means for the conclusions should be sufficient. Even if the distribution is bimodal, it doesn't invalidate the whole article, it just makes the interpretation less clear. But the editor should weigh in.

To address point #3, and as suggested by the reviewer, we now acknowledge the uncertainty around the distributions of the grandmother years and reproductive overlap, and the implications these could have for our conclusions in the manuscript (lines 754-759).

We are pleased that our previous draft satisfied the reviewer on all other points.